# Accumulation of F-actin drives brain aging and limits healthspan in *Drosophila*

Edward T. Schmid[1], Joseph M. Schinaman[1], Naomi Liu-Abramowicz[1], Kylie S. Williams[1] & David W. Walker [1,2] ✉

The actin cytoskeleton is a key determinant of cell structure and homeostasis. However, possible tissue-specific changes to actin dynamics during aging, notably brain aging, are not understood. Here, we show that there is an age-related increase in filamentous actin (F-actin) in *Drosophila* brains, which is counteracted by prolongevity interventions. Critically, decreasing F-actin levels in aging neurons prevents age-onset cognitive decline and extends organismal healthspan. Mechanistically, we show that autophagy, a recycling process required for neuronal homeostasis, is disabled upon actin dysregulation in the aged brain. Remarkably, disrupting actin polymerization in aged animals with cytoskeletal drugs restores brain autophagy to youthful levels and reverses cellular hallmarks of brain aging. Finally, reducing F-actin levels in aging neurons slows brain aging and promotes healthspan in an autophagy-dependent manner. Our data identify excess actin polymerization as a hallmark of brain aging, which can be targeted to reverse brain aging phenotypes and prolong healthspan.

The actin cytoskeleton dictates the shape and polarity of a cell and is essential in numerous and diverse fundamental processes including cellular division, motility, phagocytosis, organelle trafficking, and signaling[1]. Actin can be found in two forms: monomeric (G-actin) and filamentous (F-actin). Assembly and disassembly of actin filaments are regulated by a large number of actin-interacting proteins[2], making maintenance of the actin cytoskeleton highly susceptible to disruption caused by aging. In fact, aging is associated with not only changes in the expression of actin genes but also disruption of actin cytoskeletal dynamics[3]. More specifically, recent studies in *C. elegans* have shown that the organization of the actin cytoskeleton deteriorates in the hypodermis, muscles, and intestines during aging[4,5]. Critically, however, the interplay between actin dynamics and neuronal aging has not been characterized in any species. Intriguingly, F-actin-rich paracrystalline inclusions (known as Hirano bodies) have been described in aging and neurodegeneration, including Alzheimer's disease and other tauopathies[6–12]. Importantly, animal models expressing mutant actin binding proteins suggest Hirano bodies are associated with impaired synaptic responses and decreased spatial working memory[13,14].

Furthermore, tau-induced neurodegeneration is associated with accumulation of F-actin and the formation of actin-rich rods in both *Drosophila* and mouse models[15]. Critically, reducing F-actin levels improves symptoms in tau- and other models of neurodegeneration in *Drosophila*[15–17]. Hence, it is apparent that excess actin polymerization and actin-rich inclusions play a direct role in neurotoxicity in the context of certain diseases of the brain[15,18,19]. However, despite considerable focus upon identifying key hallmarks of aging[20], including brain aging[21], the role of actin dynamics in the aging brain has not been characterized.

Two well-characterized hallmarks of aging are the accumulation of damaged proteins and dysfunctional mitochondrial[20]. Macroautophagy (hereafter, autophagy) is a fundamental process by which cellular waste (referred to as autophagic cargo), including nucleic acids, proteins, lipids and organelles, is isolated inside specialized vesicles called autophagosomes for recycling via lysosome-mediated degradation[22]. A growing body of evidence indicates that autophagic activity declines with age[23], including in the aged brain[24–26]. This age-related decline has also been reported in the context of mitochondrial

[1]Department of Integrative Biology and Physiology, University of California, Los Angeles, Los Angeles, CA 90095, USA. [2]Molecular Biology Institute, University of California, Los Angeles, Los Angeles, CA 90095, USA. ✉e-mail: davidwalker@ucla.edu

autophagy (mitophagy), a cargo-specific form of autophagy that degrades dysfunctional mitochondria[27–29]. Importantly, stimulating autophagy or mitophagy in aging neurons can prolong lifespan in model organisms[20,29–32]. Hence, there is an emerging understanding that disabled autophagy/mitophagy contributes to brain aging and, thereby, limits lifespan[20,23,30]. The age-associated dysregulation of autophagy has been demonstrated by the accumulation of autophagosomes, possibly due to impaired lysosomal fusion and/or degradation, yet the underlying mechanisms are not understood[23]. Actin cytoskeleton dynamics play important roles throughout the various steps of autophagy[33]. Indeed, studies in cell culture have shown that actin dynamics regulates lysosome-autophagosome fusion[34,35], including by promoting autophagosome trafficking to the lysosome[36]. Interestingly, recent work has shown that excess F-actin stabilization can disrupt autophagic activity in a *Drosophila* Parkinson's disease model of α-synuclein neurotoxicity[37]. However, the interplay between actin dynamics, autophagy and brain aging remains unexplored.

In this study, we set out to examine the role of actin dynamics in brain aging. Using the *Drosophila* model, we have identified a striking increase in F-actin in the aged brain concomitant with the formation of F-actin-rich rods not present in young animals. Furthermore, we have found that F-actin levels in the brain correlate with the health of aged animals. Flies undergoing dietary restriction or treated with rapamycin, two evolutionarily conserved approaches to lifespan extension[38,39], both show a reduction in F-actin in aged brains. To establish causal relationships, we have identified multiple interventions targeting neuronal actin dynamics that can slow brain aging and prolong healthspan. More specifically, we show that adult-onset, neuron-specific inhibition of *Fhos* (*Formin homology 2 domain containing ortholog*), a FHOD class formin that nucleates actin filaments[40], improves cognitive function in aged flies and dramatically improves multiple markers of organismal healthspan. Using both genetic and pharmacological approaches, we show that excess F-actin polymerization leads to impaired autophagic activity and the accumulation of dysfunctional mitochondria in the aged brain. Remarkably, we show that treating aged animals with cytoskeletal drugs, to disrupt actin polymerization, can reverse age-onset impairments in brain autophagy and improve cognitive performance. Finally, we show that improvements in autophagy in the aged brain are necessary for the beneficial effects of neuronal F-actin modulation. Together, our findings reveal neuronal dysregulation of actin dynamics as a hallmark of aging, which can be targeted to restore autophagic activity, improve brain function and prolong healthspan.

## Results

### F-actin accumulates in aged *Drosophila* brains and correlates with health

Recent work has shown that non-neuronal tissues *in C. elegans* experience actin filament destabilization and a decline in cytoskeletal integrity with age[5]. However, little is known about actin dynamics in the brains of aging organisms. Using the *Drosophila* model, we examined actin in the brains of naturally aging animals. We began by comparing brains from wild type flies collected from young animals (10 days post eclosion) to those isolated from middle-age (30 days post eclosion) and late-age (45 days post eclosion) flies by immunofluorescent microscopy (IF), using phalloidin to stain for F-actin[15,17,37,41]. Remarkably, we detected a significant increase in total F-actin levels in brains with age (Fig. 1a, b). It appeared that the optic lobes were particularly sensitive to age-related changes to brain F-actin (Fig. 1a). At increased magnification of the optic lobes, we observed F-actin-rich rod-like structures in aged brains that were absent in young brains (Fig. 1c). Interestingly, both cytoplasmic actin isoforms expressed in *Drosophila* neurons, *Act5c* and *Act42a*[42], were observed to increase transcriptionally by quantitative polymerase chain reaction (qPCR) in aged head samples when compared to two other genes considered to

be expressed at a steady state: *GAPDH* and *RPL32* (Supplementary Fig. 1a, b). Although actin genes are widely considered to be 'housekeeping genes', these findings argue that actin expression is dynamic with age in *Drosophila* brains.

To corroborate and extend our findings, we used reporter lines to express Act5c and Act42a tagged with green fluorescent protein (GFP) in neurons. Act5c-GFP distribution displayed remarkable colocalization with age-associated F-actin-rich rods labeled by phalloidin (Fig. 1d). Neuronal expression of Act42a-GFP showed actin-rich structures in aging brains that were absent in young brains (Supplementary Fig. 1c, d). Interestingly, the distribution of Act42a-GFP in aging *Drosophila* brains showed a pattern distinct from that of Act5c-GFP and phalloidin, suggesting the possibility of a different distribution of this actin isoform in neurons. Additionally, labeling brains with anti-actin antibody revealed a greater overall actin intensity in aged brains compared to young brains (Supplementary Fig. 1e, f). Increased magnification in the brain revealed a dramatic difference in F-actin-rich rods with age (Fig. 1e, f). Furthermore, both middle-age and later-aged fly head protein homogenates had significantly more F-actin compared to young controls as detected by enzyme-linked immunosorbent assays (ELISAs) (Fig. 1g). No change in total head protein levels with age was detected by Bradford assay (Supplementary Fig. 1g.).

To further define the cellular localization of the age-related changes to F-actin observed in the optic lobe, we labeled brains with antibodies targeting N-cadherin, a cell adhesion molecule associated with axon and dendrite outgrowth and synapse formation, abundant in the neuropil. We detected the presence of N-cadherin where we found the strongest age-related changes to brain F-actin (Supplementary Fig. 1h). Although the *Drosophila* brain is made up of approximately 90% neurons, glia comprise a sizeable population of brain cells that play an essential role in supporting neuronal function and forming the blood-brain barrier. To test if the region in the optic lobe in which we observed the dramatic age-related changes to F-actin was specifically associated with neurons or glia, we expressed a cell membrane-targeted GFP reporter (UAS-mCD8GFP) with GAL4 drivers for neurons and the five sub-types of glia (perineural glia, subperineural glia, cortex glia, astrocyte-like glia, and ensheathing glia)[43]. Here, the neuronal reporter shared the closest distribution pattern with the F-actin-rich rods (Supplementary Fig. 1i). The pharmacological reagents cytochalasin D and latrunculin A are commonly used to depolymerize actin filaments[44,45]. Remarkably, feeding aged flies cytochalasin D (Supplementary Fig. j, k) or latrunculin A (Supplementary Fig. l, m) for one week was sufficient to ablate the age-associated accumulation of F-actin-rich rods in brains. Cumulatively, these observations support a model in which F-actin polymerization increases in *Drosophila* brains with age.

To assess if F-actin polymerization in aged brains was reflective of aging health or if it occurred universally with chronological time, we evaluated flies from two widely studied lifespan extension strategies. Dietary restriction (DR) and/or protein restriction is an evolutionarily conserved approach to slow aging and promote longevity[46]. Flies fed a low-protein diet had a significantly longer lifespan compared to those provided a high-protein diet (Fig. 1h). Using IF, we observed F-actin-rich rods in the brains of flies on a rich diet at young middle-age (21 days post eclosion) that were absent in the brains of flies undergoing DR (Fig. 1i, j). We next tested the effect of treating flies with rapamycin, a small molecule that has also been shown to prolong lifespan in evolutionarily diverse species via inhibition of mTORC1[39]. Consistent with previous observations[47,48], feeding flies rapamycin significantly extended lifespan compared to vehicle-fed controls (Fig. 1k). Furthermore, aged flies that were fed rapamycin had significantly fewer F-actin-rich rods in the brain compared to age-matched controls (Fig. 1l, m). Together, these findings suggest that age-associated F-actin polymerization in *Drosophila* brains reflects aging health and can be counteracted by prolongevity strategies.

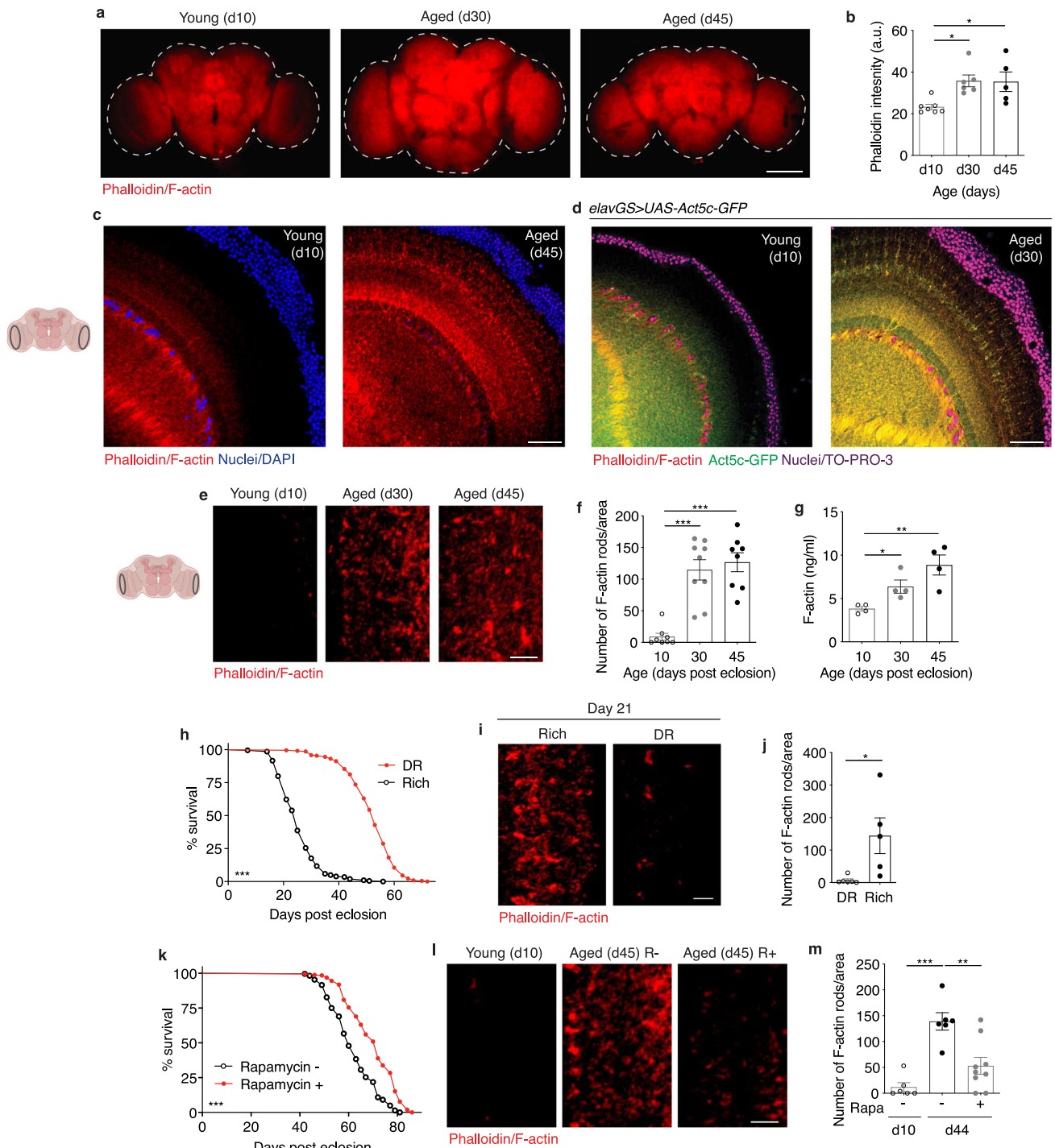

## Reducing neuronal F-actin levels during aging prolongs healthspan

Since our observations found a correlation between aging and F-actin accumulation in the brain, we next decided to test if targeting neuronal F-actin polymerization genetically could affect organismal healthspan. We screened several genes related to actin and actin stabilization, including actin isoforms, actin-binding proteins (ABPs), and actin assembly factors, and assessed changes to age-related F-actin polymerization in the brain and to organismal lifespan. We found that neuronal knockdown of *Formin homology 2 domain containing ortholog* (*Fhos*), the *Drosophila* homolog of the FHOD sub-family of formins, had the most robust effect on these parameters. *Fhos* promotes actin nucleation and is important for filament assembly[40]. Using the pan-

neuronal *Elav*-Gene-Switch (*elav*GS) driver line[49], we expressed *UAS-Fhos-RNAi* in the adult fly neurons upon administration of the inducing agent RU486 in food. This system allows for cell-type-specific induction of genetic constructs in a time- and dose-dependent manner. Flies in each experiment come from the same parental crosses and undergo identical developmental conditions. Phalloidin staining in aged brains revealed that neuronal *Fhos-RNAi* induction abrogated the F-actin-rich rods observed in aged control brains (Fig. 2a, b). To deepen our understanding of F-actin dynamics in aging brains, and to determine if neuronal *Fhos* knockdown had universal effects throughout the brain, we next imaged the antennal lobe and mushroom body by IF. Similar to the optic lobe, we found an age-associated increase in F-actin in the antennal lobe (Supplementary Fig. 2a, b). Furthermore, neuronal *Fhos*

**Fig. 1 | F-actin accumulates in aged *Drosophila* brains and correlates with health. a** Immunostaining of brains at 10x magnification from young (10-day-old) and aged (30-day-old and 45-day-old, as indicated) Canton S flies, showing F-actin fluorescence intensity (red channel, phalloidin). Scale bar is 100 μm. **b** Quantification of mean phalloidin fluorescence intensity in brains as shown in **a**. d10 $n = 7$, d30 $n = 6$, and d45 $n = 5$ flies, as indicated. *$p$ (d10 vs. d30) = 0.0158, *$p$ (d10 vs. d45) = 0.0263; one-way ANOVA, Tukey's multiple comparisons test. **c** Immunostaining of brains at 63x magnification from young (10-day-old) and aged (45-day-old) flies, showing F-actin (red channel, phalloidin) and nuclear DNA (blue channel, DAPI). These findings were repeated in more than 15 independent experiments. Scale bar is 20 μm. Accompanying diagram indicates brain region where imaging was conducted. **d** Immunostaining of brains from young (10-day-old) and aged (30-day-old) *elavGS > UAS-Act5c-GFP* flies, showing F-actin (red channel, phalloidin), Act5c-GFP (green channel), and nuclear DNA (blue channel, To-Pro-3). These findings were repeated in 3 independent experiments. Scale bar is 20 μm. **e** Immunostaining of brains at 63x magnification from young (10-day-old) and aged (30-day-old and 45-day-old, as indicated) Canton S flies, showing F-actin-rich rods (red channel, phalloidin). Accompanying diagram indicates brain region where imaging was conducted. **f** Quantification of F-actin-rich rods per 1 mm² area of brain optic lobes as shown in **e**. d10 $n = 8$, d30 $n = 9$, d45 $n = 8$ flies, as indicated. ***$p$ (d10 vs d30 and d10 vs d45) < 0.0001; one-way ANOVA,

Tukey's multiple comparisons test. **g** Quantification of F-actin protein by ELISA using head homogenates from young (10-day-old) and aged (30-day-old and 45-day-old, as indicated) Canton S flies. $n = 4$ homogenates generated with 5 brains each per condition. *$p$ = 0.0192, **$p$ = 0.0053, unpaired two-tailed t-tests. **h** Survival curves of Canton S flies given a rich diet (5.0% yeast extract) versus those undergoing dietary restriction (DR, 0.5% yeast extract) from day 4 post eclosion onwards. ***$p$ = 0.001; log-rank test; $n > 140$ flies. **i** Immunostaining of brains from Canton S flies aged day 21 post eclosion provided a rich or restricted (DR) diet, as in (h), showing F-actin-rich rods (red channel, phalloidin). Scale bar is 5 μm. **j** Quantification of F-actin-rich rods by phalloidin stain per 1 mm² area of brains as shown in **i**. DR $n = 6$, Rich $n = 5$ flies, as indicated. *$p$ = 0.0225, unpaired two-tailed t-test. **k** Survival curves of white Dahomey flies given 10 μM rapamycin or vehicle from day 4 post eclosion onwards. ***$p$ < 0.0001; log-rank test; $n > 250$ flies per condition. **l** Immunostaining of brains from young (10-day-old) and aged (45-day-old) white Dahomey flies given 10 μM rapamycin or vehicle, showing F-actin-rich rods (red channel, phalloidin). Scale bar is 5 μm. **m** Quantification of F-actin-rich rods by phalloidin stain per 1 mm² area of brains as shown in **l**. d10 $n = 6$, d44 Rapa - = 6, d44 Rapa + = 9 flies, as indicated. **$p$ = 0.0023., ***$p$ = 0.0001; one-way ANOVA, Tukey's multiple comparisons test. Data are presented as scatter plots overlaying mean values +/- SEM.

knockdown resulted in F-actin levels in aged antennal lobes comparable to young brains (Supplementary Fig. 2a, b). However, we found no change in F-actin intensity with age or with neuronal *Fhos* knockdown in the mushroom body (Supplementary Fig. 2c, d). These findings indicate that not all brain regions may undergo the same changes to actin dynamics with age or be affected by genetic interventions targeting neuronal age-associated F-actin accumulation.

Actin plays an essential role in neuron polarity and the organization of synapses and, consequently, neuronal function[50]. To assess physiological brain function, we tested associative learning and memory using olfaction aversion training[51]. Briefly, flies were conditioned to associate a neutral odor (3-octanol, OCT) with a series of electric shocks. After one hour of rest, they were placed in a T-maze and allowed to choose between OCT and a second neutral odor (4-methylcyclohexanol). Young flies avoided the shock-associated OCT significantly more than aged flies (Fig. 2c). Furthermore, aged flies expressing *Fhos-RNAi* in neurons showed a significantly better memory recall response than uninduced age-matched controls (Fig. 2c). Remarkably, treating aged flies for one week with the actin destabilization drug cytochalasin D also significantly improved associative learning and memory (Fig. 2d). Hence, reducing F-actin polymerization in aged brains, either genetically or pharmacologically, improved learning and memory in aged flies.

Changes in food intake can extend lifespan and delay age-related brain F-actin polymerization (Fig. 1h, i, j). To test if neuronal expression of *Fhos-RNAi* in adult flies affected feeding behavior, we performed the consumption-excretion ("Con-Ex") feeding assay[52]. However, we observed no differences in food consumption and excretion (Fig. 2e). Insulin/IGF-1 signaling (IIS) has been shown to be important for organismal health and to influence lifespan[53]. To assess if neuronal *Fhos* knockdown directly affects IIS, we examined *Drosophila* insulin-like peptide (DILP) levels in the brain. We observed no significant change to DILP2 levels in the insulin-producing cells (IPCs) of *elavGS > UAS-Fhos-RNAi* flies after one week of knockdown compared to controls (Supplementary Fig. 2e, f). Additionally, we tested if the translational regulator *4E-BP*, a transcriptional target of *Drosophila* FOXO (dFOXO) that is induced with the repression of IIS, was affected by neuronal *Fhos* knockdown. In agreement with no change in DILP2 levels (Supplementary Fig. 2e, f), we found no change to *4E-BP* expression in heads, thoraces, or guts after one week of neuronal *Fhos-RNAi* expression (Supplementary Fig. 2g, h, i). Cumulatively, it appears that neuronal knockdown of *Fhos* does not directly affect *Drosophila* feeding behavior or IIS.

With neuronal *Fhos*-RNAi expression correlating with a reduction in age-associated F-actin-rich rods in the brain and an improvement in memory, we next decided to test if organismal lifespan and additional parameters of health were also improved. Remarkably, neuronal *Fhos* knockdown in *elavGS > UAS-Fhos-RNAi* flies showed a dramatic increase in lifespan compared to controls (Fig. 2f). This effect was validated using elav-Gal4 and the tub-Gal80-ts system (Supplementary Fig. 2j). We also tested if *Fhos* knockdown in glia, the other primary cell population of the brain, had an effect on lifespan. However, we found no change in lifespan with an inducible glia-specific driver (Supplementary Fig. 2k). Interestingly, several additional genes targeting actin dynamics in neurons conferred improvements to lifespan. Midlife neuronal knockdown of *Act5c* significantly extended organismal lifespan (Supplementary Fig. 2l), while knockdown of *Act42a* had no discernable effect (Supplementary Fig. 2m). Neuronal overexpression of *Twinstar* (*tsr*), the *Drosophila* homolog of *cofilin/ADF* (actin depolymerization factor) that plays a major role in regulating actin cytoskeletal dynamics[54], also extended *Drosophila* lifespan (Supplementary Fig. 2n). Furthermore, neuronal overexpression of *Gelsolin*, an actin binding protein with filament capping and severing functions, also extended lifespan (Supplementary Fig. 2o). Conversely, overexpressing *Act5c* and *Act42a* in neurons significantly reduced fly lifespan (Supplementary Fig. 2p, q). To compare the cellular effect of neuronal *Fhos* knockdown on F-actin-rich rods (Fig. 2a, b) with another actin-targeting line, we imaged optic lobes of *elavGS > UAS-Act5c-RNAi* flies. Neuronal *Act5c-RNAi* expression also resulted in a reduction in age-associated F-actin-rich rods (Supplementary Fig. 2r, s). Taken together, we found that genetically targeting multiple actin genes in neurons associated with filament assembly and turnover reduced the presence of age-associated F-actin-rich rods in brains and extended organismal lifespan.

Since neuronal knockdown of *Fhos* conferred the greatest extension in *Drosophila* lifespan of the screened actin dynamics genes, we further characterized changes to healthspan in these flies. In agreement with improved lifespan, aged *elavGS > UAS-Fhos-RNAi* flies treated with RU486 showed improved locomotor activity (Fig. 2g) and climbing endurance (Fig. 2h) compared to vehicle-fed controls. Additionally, neuronal *Fhos-RNAi* induction resulted in an increase in spontaneous daytime activity with no detectable nighttime restlessness in aged flies (Fig. 2i, j). Intestinal barrier dysfunction is an evolutionarily-conserved characteristic of aging associated with systemic inflammation, frailty, and mortality[55]. To assess if neuronal knockdown of *Fhos* could prolong intestinal integrity, we performed

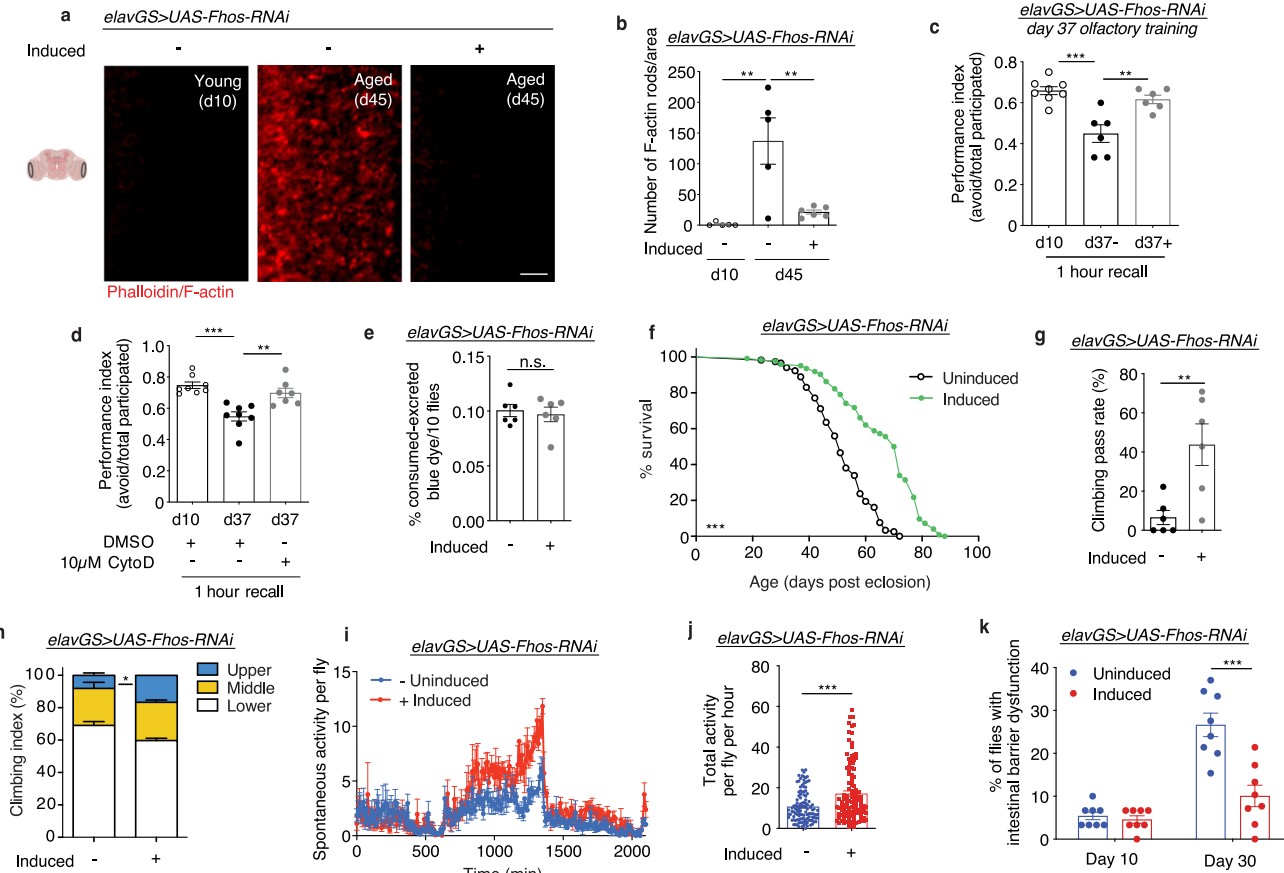

**Fig. 2 | Neuronal knockdown of *Fhos* reduces age-associated F-actin-rich rods and extends healthspan. a** Immunostaining of brains at 63x magnification from young (10-day-old) and aged (45-day-old) *elavGS > UAS-Fhos-RNAi* flies with or without RU486-mediated transgene expression from day 5 onward, showing F-actin-rich rods (red channel, phalloidin). The scale bar is 5 μm. Accompanying diagram indicates brain region where imaging was conducted. **b** Quantification of F-actin-rich rods per 1 mm² area of brain optic lobes as shown in **a**. d10 *n* = 5, d45 uninduced = 5, d45 induced = 6 flies, as indicated. **\*\****p* (d10 vs. aged uninduced) = 0.0015, **\*\****p* (aged uninduced vs. aged induced) = 0.0037, one-way ANOVA, Tukey's multiple comparisons test. **c** Performance index in olfactory aversion training for 37-day-old *elavGS > UAS-Fhos-RNAi* flies with or without RU486-mediated transgene expression from day 5 onward, assessed by the number of flies avoiding a shock-associated odor versus the total number of flies participating in the assay. d10 *n* = 8, d37 uninduced = 6, d37 induced = 6 flies, as indicated. **\*\****p* = 0.0026, **\*\*\****p* = 0.0001, one-way ANOVA, Tukey's multiple comparisons test. **d** Performance index in olfactory aversion training for 37-day-old Canton S flies given vehicle (DMSO) or 10 μM cytochalasin D as indicated from days 30-37 post eclosion, assessed by the number of flies avoiding a shock-associated odor versus the total number of flies participating in the assay. d10 *n* = 8, d37 DMSO *n* = 8, d37 CytoD *n* = 7 flies, as indicated. **\*\****p* = 0.0023, **\*\*\****p* = 0.0001, one-way ANOVA, Tukey's multiple comparisons test. **e** Con-ex feeding assay of 10-day-old *elavGS > UAS-Fhos-RNAi* flies with or without RU486-mediated transgene expression from day 5 onward. *n* = 6 vials of 10

flies per condition. ns = nonsignificant, unpaired two-tailed t-test. **f** Survival curve of *elavGS > UAS-Fhos-RNAi* flies with or without RU486-mediated transgene expression from day 5 onward. **\*\*\****p* < 0.0001, log-rank test. *n* = 118 uninduced and 124 induced biologically independent animals. **g** Climbing pass rate of 58-day-old *elavGS > UAS-Fhos-RNAi* flies with or without RU486-mediated transgene expression from day 5 onward. *n* = 180 biologically independent animals per condition measured in groups of 30. **\*\****p* = 0.0079, unpaired two-tailed t-test. **h** Climbing index as a measure of endurance of 45-day-old *elavGS > UAS-Fhos-RNAi* flies with or without RU486-mediated transgene expression from day 5 onward. n = 4 replicates of uninduced groups and 4 replicates of induced groups with 100 biologically independent animals per replicate. **\****p* = 0.0267, unpaired two-tailed t-test. **i** Spontaneous physical activity of 44-day-old *elavGS > UAS-Fhos-RNAi* flies with or without RU486-mediated transgene expression from day 5 onward. *n* = 3 vials of 10 flies per condition. **j** Quantification of total activity per fly per hour from spontaneous activity graphs in **i**. *n* = 3 vials of 10 biologically independent animals per condition. **\*\*\****p* < 0.0001, unpaired two-tailed t-test. **k** Intestinal integrity during aging of *elavGS > UAS-Fhos-RNAi* flies with or without RU486-mediated transgene expression from day 5 onward. *n* = 8 vials of 30 biologically independent animals per vial on day 10. **\*\*\****p* < 0.0001; two-way ANOVA/Šídák's multiple comparisons test. RU486 or vehicle was provided in the media at a concentration of 50 ug/ml in the indicated treatment groups. Data are presented as scatter plots overlaying mean values +/− SEM.

the 'Smurf assay'[56,57]. In agreement with prolonged lifespan and improved parameters of aging health, we observed a delay in gut leakiness in aged flies expressing *Fhos-RNAi* in neurons (Fig. 2k).

To confirm that the effects of neuronal modulation of *Fhos, Act5c, Tsr*, and *Gelsolin* were not an artifact of construct expression or RU486 administration, we generated flies expressing double-stranded RNA of *GFP* in neurons (*elavGS > UAS-dsRNA-GFP*). Induction of *dsRNA-GFP* did not affect feeding behavior (Supplementary Fig. 2t), age-associated brain F-actin polymerization (Supplementary Fig. 2u, v), or memory and learning (Supplementary Fig. 2w). Furthermore, providing RU486 to *elavGS > UAS-dsRNA-GFP* flies did not extend organismal lifespan

(Supplementary Fig. 2x). Together, these findings validate that neuronal knockdown of the actin-targeting genes *Fhos* and *Act5c*, as well as overexpression of *Tsr* and *Gelsolin*, can significantly delay parameters of aging and extend organismal healthspan.

## Age-associated neuronal F-actin polymerization impairs brain autophagy

Actin dynamics are essential in the biogenesis and transportation of most cellular vesicles, including autophagosomes[33]. Disabled autophagy is considered a primary hallmark of aging, resulting in impaired proteostasis and decreased organelle turnover[20,23,30,58]. Furthermore,

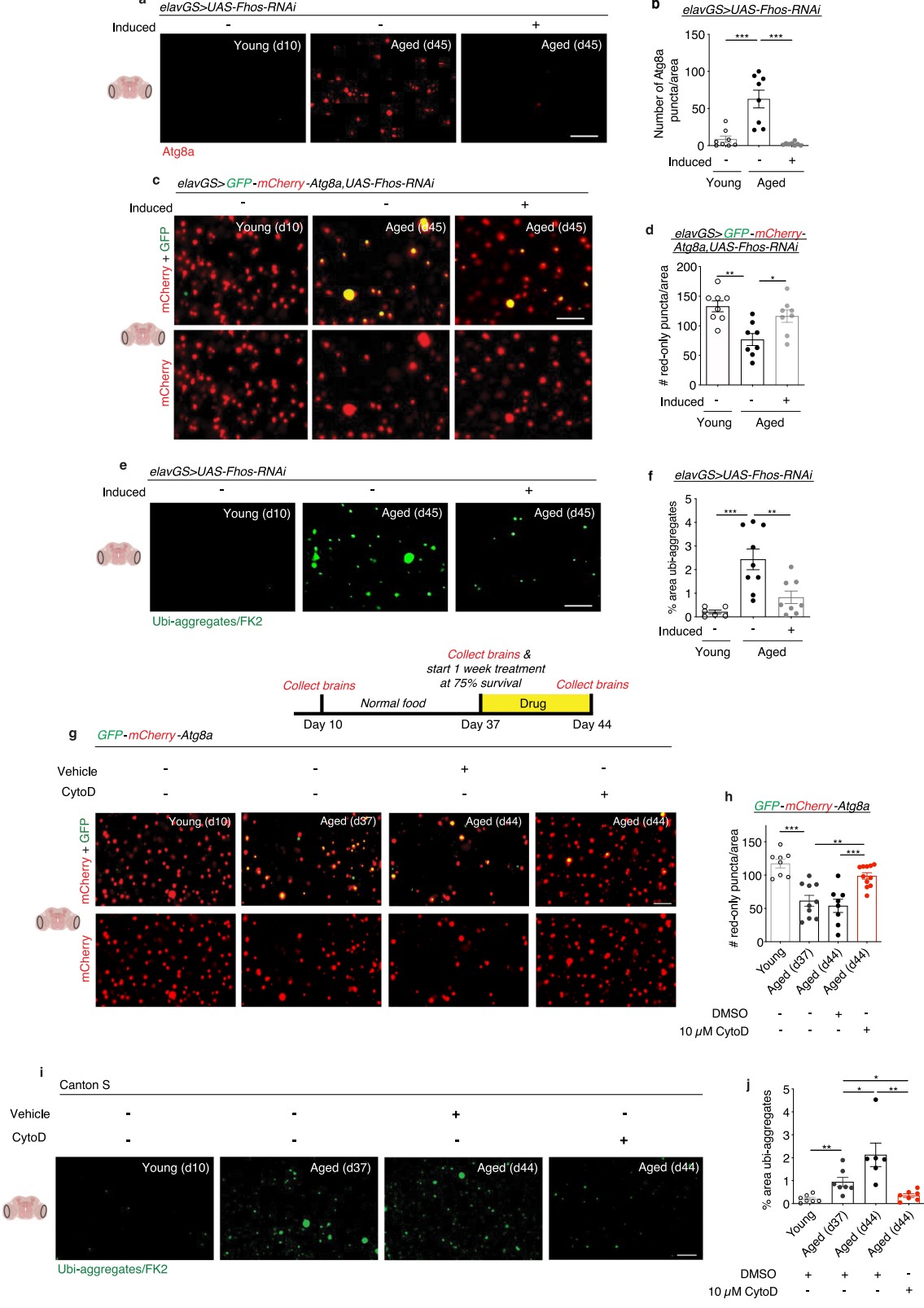

autophagy and vesicular trafficking defects have been identified in neurodegenerative diseases, including Alzheimer's disease, Parkinson's disease, and Huntington's disease[37,59]. With the changes observed to actin dynamics in the aging brain (Fig. 1), we next sought to evaluate if interventions targeting age-associated F-actin polymerization would affect brain autophagy. Using endogenous LC3/Atg8 as a marker of autophagy, we found an age-related increase in autophagosomes

reflective of reduced autophagic flux[60,61] (Fig. 3a, b). Neuronal-specific knockdown of actin nucleation factor *Fhos* resulted in a dramatic reduction in Atg8a accumulation, closely resembling what was observed in young brains (Fig. 3a, b). To further evaluate autophagic activity in the aging brain, we used a reporter line expressing GFP-mCherry-Atg8a ("Atg8a-tandem") ubiquitously under the control of the endogenous Atg8 promoter[62]. As autophagosomes fuse with

**Fig. 3 | F-actin stabilization disables autophagy in the aged *Drosophila* brain. a** Immunostaining of brains at 63x magnification from young (10-day-old) and aged (45-day-old) *elavGS > UAS-Fhos-RNAi* flies with or without RU486-mediated transgene induction from day 5 onward, showing Atg8a (red channel, anti-Atg8a). Scale bar is 5 μm. Accompanying diagrams indicate brain region where imaging was conducted. **b** Quantification of Atg8a puncta in the brain as shown in **a**. *n* = 8 biologically independent animals, as indicated. ***p < 0.0001 (young vs. aged uninduced), ***p < 0.0001 (aged uninduced vs. aged induced); one-way ANOVA, Tukey's multiple comparisons test. **c** GFP-mCherry-Atg8a in brains at 63x magnification from 45-day-old *elavGS > GFP-mCherry-Atg8a,UAS-Fhos-RNAi* flies. Images show mCherry and merged GFP-mCherry channels. The scale bar is 5 μm. **d** Quantification of the number of autolysosomes (red-only puncta) in brains as shown in **c**. *n* = 8 biologically independent animals per condition. *p = 0.0263; **p = 0.0018; one-way ANOVA, Tukey's multiple comparisons test. **e** Immunostaining of brains at 63x magnification from young (10-day-old) and aged (45-day-old) *elavGS > UAS-Fhos-RNAi* flies with or without RU486-mediated transgene induction from day 5 onward, showing poly-ubiquitinated aggregates (green channel, anti-FK2). Scale bar is 5 μm.
**f** Quantification of % area of polyubiquitinated aggregates in brains as shown in **e**. Young *n* = 6, aged uninduced *n* = 9, aged induced *n* = 8 flies, as indicated.

**p = 0.0060, ***p = 0.0006; one-way ANOVA/Tukey's multiple comparisons test. **g** GFP-mCherry-Atg8a in brains from 10-day-old, 37-day-old, and 44-day-old Canton S flies given vehicle (DMSO) or 10 μM cytochalasin D as indicated from days 37-44 post eclosion. Images show mCherry and merged GFP-mCherry channels. The scale bar is 5 μm. **h** Quantification of the number of autolysosomes (red-only puncta) in brains as shown in **g**. Young n = 8, aged (d37) DMSO *n* = 10, aged (d44) DMSO *n* = 8, aged (d44) CytoD *n* = 11 biologically independent animals, as indicated. ***p (young vs. aged d37) < 0.0001; **p (aged d37 vs. aged d44 CytoD) = 0.0030; ***p (aged d44 DMSO vs. aged d44 CytoD) = 0.0007, one-way ANOVA, Tukey's multiple comparisons test. **i** Immunostaining of brains from 10-day-old, 37-day-old, and 44-day-old Canton S flies given vehicle (DMSO) or 10 μM cytochalasin D as indicated from days 37-44 post eclosion, showing polyubiquitinated aggregates (green channel, anti-FK2). The scale bar is 5 μm. **j** Quantification of % area of polyubiquitinated aggregates in brains as shown in **i**. Young *n* = 7, aged (d37) DMSO *n* = 7, aged (d44) DMSO *n* = 6, aged (d44) CytoD *n* = 7 biologically independent animals, as indicated. **p (young vs. aged d37) = 0.0042; *p (aged d37 vs. aged d44 DMSO) = 0.0441, *p (aged d37 vs. aged d44 CytoD) = 0.0172, **p (aged d44 DMSO vs. aged d44 CytoD) = 0.0035; unpaired two-tailed t-tests. Data are presented as scatter plots overlaying mean values +/− SEM.

lysosomes, GFP signal on the Atg8a tandem protein is quenched due to its sensitivity to low pH. Remaining mCherry-only foci indicate autolysosomal activity. When investigating the brains of young flies expressing Atg8a-tandem, we observed a striking density of red-only puncta and a near absence of green signal. Conversely, aged brains showed a mixture of yellow and red-only puncta, with significantly fewer mCherry foci indicative of fewer autolysosomes (Fig. 3c, d). These findings suggest that young brains display extensive autophagic activity that becomes stalled with age. When *Fhos-RNAi* was expressed in aging neurons of adult Atg8a-tandem flies, we observed fewer yellow puncta and more red-only puncta in brains compared to aged controls, indicating increased levels of autophagy (Fig. 3c, d).

To complement our findings with Atg8, we tested additional readouts of protein homeostasis (proteostasis) and autophagy. A decline in proteostasis is another major cellular hallmark of aging[20,58], and it has been well characterized that aged tissues in *Drosophila* accumulate aggregates of ubiquinated proteins[29,31,32,47,60,63,64]. We observed that neuronal *Fhos-RNAi* induction significantly reduced age-associated protein aggregates in the brain (Fig. 3e, f). Ubiquinated proteins can be targeted for autophagic degradation by the adaptor protein p62/SQSTM1, with an accumulation of p62 indicating reduced turnover and breakdown similar to Atg8a accumulation[55,60]. Aged brains showed significantly more p62 puncta compared to young brains, and neuronal knockdown of *Fhos* reduced the accumulation of p62 in aged brains (Supplementary Fig. 3a, b). Conversely, overexpression of the actin isoform *Act42a* in neurons using *elavGS > UAS-Act42aGFP* flies resulted in an early accumulation of p62 puncta compared to uninduced controls (Supplementary Fig. 3c, d). Furthermore, pharmacological polymerization of actin by treating flies with jasplakinolide led to an early formation of F-actin-rich rods and accumulation of p62 puncta in brains compared to age-matched vehicle-fed controls (Supplementary Fig. 3e, f, g). These findings indicate a direct relationship between excessive F-actin stabilization and stalled autophagy in the brain.

Next, we sought to test if a pharmacological intervention reducing actin polymerization could improve readouts of brain autophagy. We had earlier found that one-week treatment at midlife with cytochalasin D was sufficient to abrogate age-associated brain F-actin polymerization (Supplementary Fig. 1j, k). Here, we tested the effect of cytochalasin D on flies expressing the Atg8a-tandem reporter. In a given cohort, brains were collected from young flies (10 days post eclosion), aged flies before treatment (37 days post eclosion), and aged flies following 1 week of treatment with cytochalasin D or vehicle from day 37 to day 44 post eclosion. Aged brains collected at day 37 and day 44 fed vehicle showed significantly fewer red-only Atg8a-tandem puncta

compared to young brains, indicating a decline in autophagic activity. Remarkably, 1 week of cytochalasin D treatment in aged animals significantly increased red-only Atg8a-tandem foci compared to both day 44 and, critically, day 37 brains (Fig. 3g, h). These findings indicate a reversal in age-related decline in autophagic activity upon treating animals with a pharmacological inhibitor of actin polymerization. We next followed the same drug treatment paradigm while investigating proteostasis. Consistent with what was observed using the Atg8a-tandem reporter, treatment of wild-type flies for 1 week with cytochalasin D was sufficient to reverse the age-related accumulation of ubiquinated proteins in the brain (Fig. 3i, j). These findings imply that therapeutic targeting of age-associated actin polymerization, in aged animals, may reverse both cellular hallmarks of brain aging and improve brain function.

## Age-associated neuronal F-actin polymerization impairs brain mitophagy

One major role of autophagy is to mediate the turnover and clearance of damaged or unnecessary mitochondria[65]. Accordingly, we next sought to understand if age-associated F-actin polymerization in the brain interfered with mitophagy and mitochondrial function. To visualize mitophagic flux, we used the mito-QC reporter line that encodes a tandem GFP-mCherry fusion protein targeted to the outer mitochondria membrane[66]. Mitochondria degraded in acidic lysosomes (mitolysosomes) appear as mCherry-only puncta as GFP is quenched. Consistent with our observations in autophagy, we found significantly more red-only foci in the aged brains of flies with neuronal knockdown of the F-actin nucleation factor *Fhos* compared to age-matched controls (Fig. 4a, b). In agreement, midlife treatment of flies with cytochalasin D resulted in significantly more mitolysosomes in the brain compared to vehicle-fed controls (Supplementary Fig. 4a, b). Consistent with reduced clearance of mitochondria, brains from aged flies showed an accumulation of mitochondrial content compared to young adult controls (Fig. 4c, d) as previously described[29]. Neuronal knockdown of *Fhos* resulted in a significant reduction in mitochondrial content to an amount similar to that detected in young brains (Fig. 4c, d). Additionally, aged brains showed significantly more non-nuclear dsDNA compared to young controls, and neuronal expression of *Fhos-RNAi* significantly reduced this accumulation (Supplementary Fig. 4c, d). We next evaluated if neuronal knockdown of *Fhos* could have cell non-autonomous and/or systemic effects during aging. The *Drosophila* indirect flight muscle is a mitochondria-dense tissue that is well-suited for studies of mitochondrial homeostasis. As previously reported, mitochondria size increased in aging indirect flight muscles compared to young controls[60,64,67], while neuronal induction of *Fhos-*

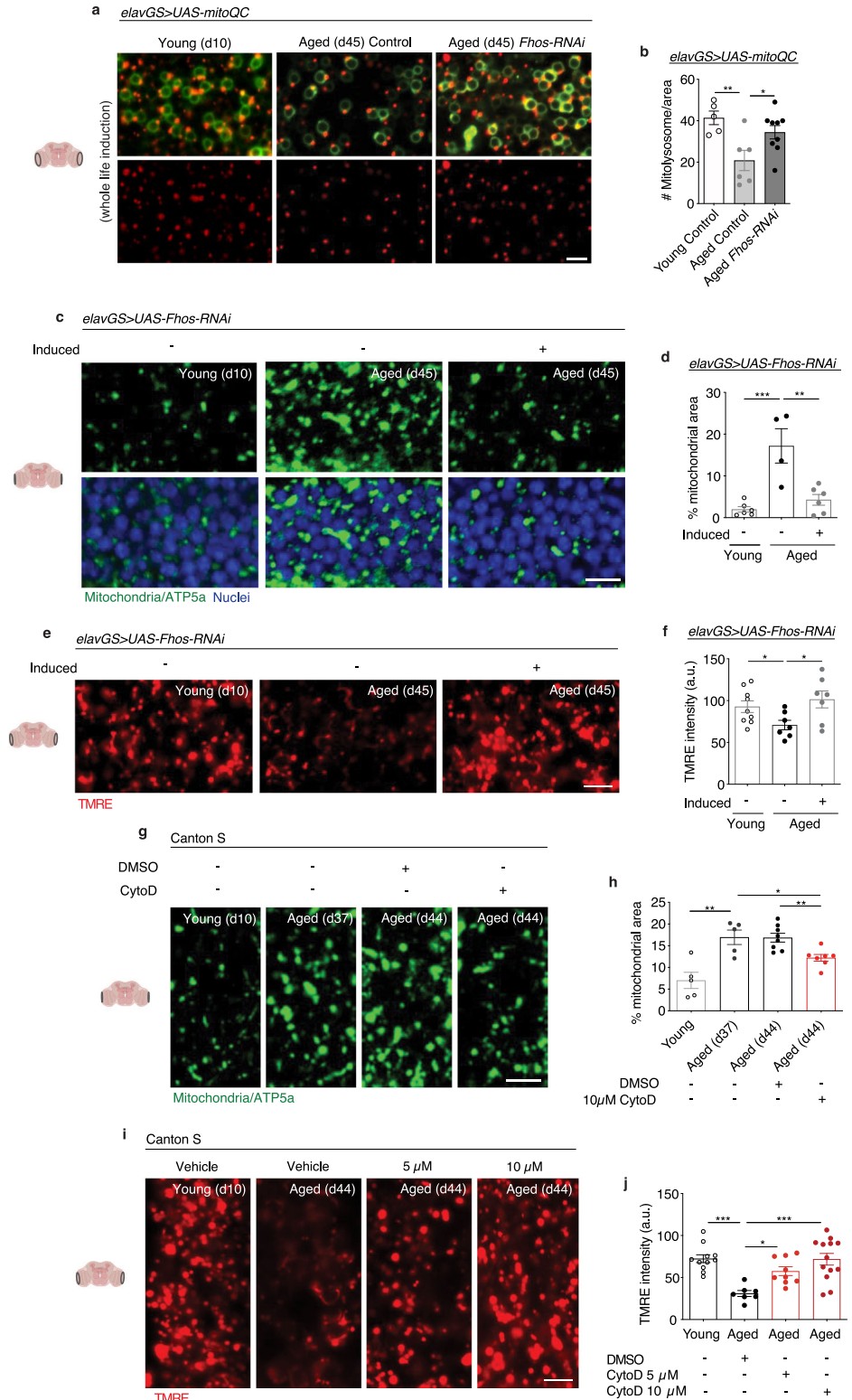

*RNAi* resulted in smaller mitochondria more closely resembling young muscle (Supplementary Fig. 4e, f). We additionally found that neuronal induction of *Fhos-RNAi* resulted in significantly reduced age-associated protein aggregates in indirect flight muscle (Supplementary Fig. 4g). In agreement with neuronal knockdown of *Fhos*, midlife neuronal knockdown of *Act5c* also resulted in reduced mitochondrial content in aged brains (Supplementary Fig. 4j, k). Control *elavGS > UAS-dsRNA-GFP* flies showed no difference in brain mitochondrial content in aged flies treated with RU486 or vehicle (Supplementary Fig. 4l, m). To

assess mitochondrial function, we examined mitochondrial membrane potential using the potentiometric dye tetramethylrhodamine ethyl ester (TMRE). Aged brains showed a significant reduction in TMRE intensity compared to young controls, while targeting neuronal F-actin polymerization via *Fhos* knockdown resulted in increased mitochondrial membrane potential (Fig. 4e, f). Furthermore, neuronal expression of *Fhos-RNAi* also increased muscle TMRE intensity compared to age-matched controls and, remarkably, even young controls (Supplementary Fig. 4h, i). These data suggest that genetic targeting of F-actin

**Fig. 4 | F-actin stabilization disables mitophagy in the aged *Drosophila* brain.**
**a** Mito-QC of brains at 63x magnification from 10-day-old and 45-day-old flies. Genotypes analyzed were *elavGS > UAS-mito-QC*, as controls, and *elavGS > UAS-mito-QC,UAS-Fhos-RNAi*. RU486-mediated transgenes were induced from day 5 onwards. Images shown of merged GFP and mCherry along with punctate mCherry-only foci (from merged images where GFP has been quenched; mitolysosomes). Scale bar is 5 μm. Accompanying diagrams indicate brain region where imaging was conducted. **b** Quantification of mitolysosomes per 4 mm² brain area as shown in **a**. Young control $n = 5$, aged control $n = 6$, aged *Fhos-RNAi* $n = 9$ biologically independent animals, as indicated. *$p = 0.0475$, **$p = 0.0089$, one-way ANOVA/Tukey's multiple comparisons test. **c** Immunostaining of brains at 63x magnification from young (10-day-old) and aged (45-day-old) *elavGS > UAS-Fhos-RNAi* flies with or without RU486-mediated transgene induction from day 5 onward, showing mitochondrial morphology (green channel, anti-ATP5a) and nuclear DNA (blue channel, stained with To-Pro-3). Scale bar is 5 μm. **d** Quantification of mitochondrial area in brain as shown in **c**. Young = 6, aged uninduced $n = 4$, aged induced $n = 6$ biologically independent animals, as indicated. **$p = 0.0018$, ***$p = 0.0004$; one-way ANOVA/Tukey's multiple comparisons test. **e** Staining of brains at 63x magnification from young (10-day-old) and aged (45-day-old) *elavGS > UAS-Fhos-RNAi* flies with or without RU486-mediated transgene induction from day 5 onward, showing TMRE fluorescence. Scale bar is 5 μm. **f** Quantification of mitochondrial membrane

potential measured by TMRE staining as shown in **e**. Young $n = 9$, aged uninduced $n = 7$, aged induced $n = 7$ biologically independent animals, as indicated. *p (d10 vs. d45 uninduced) = 0.0335, *p (d45 uninduced vs. d45 induced) = 0.0221; one-way ANOVA/Tukey's multiple comparisons test. (**g**) Immunostaining of brains from 10-day-old, 37-day-old, and 44-day-old Canton S flies given vehicle (DMSO) or 10 μM cytochalasin D as indicated from days 37-44 post eclosion, showing mitochondrial morphology (green channel, anti-ATP5a). Scale bar is 5 μm. **h** Quantification of mitochondrial area in brain as shown in **g**. Young $n = 5$, aged (d37) $n = 5$, aged (d44) DMSO $n = 8$, aged (d44) CytoD $n = 7$ biologically independent animals, as indicated. **p (young vs. aged d37) = 0.0041, *p (aged d37 vs. aged d44 CytoD) = 0.0174, **p (aged d44 DMSO vs. aged d44 CytoD) = 0.0036; unpaired two-tailed t-tests. **i** Staining of brains from young (10-day-old) and aged (45-day-old) Canton S flies given vehicle (DMSO), 5 μM, or 10 μM cytochalasin D as indicated from days 37-44 post eclosion, showing TMRE fluorescence. Scale bar is 5 μm. **j** Quantification of mitochondrial membrane potential measured by TMRE staining in brains as shown in **i**. Young $n = 11$, aged DSMO $n = 7$, aged CytoD 5 μM $n = 9$, aged CytoD 10 μM $n = 13$ biologically independent animals, as indicated. ***p (young vs. aged d44 DMSO) = 0.0003, *p (aged d44 DMSO vs. aged d44 5 μM CytoD) = 0.0353, ***p (aged d44 DMSO vs. aged d44 10 μM CytoD) = 0.0002; one-way ANOVA/Tukey's multiple comparisons test. Data are presented as scatter plots overlaying mean values +/− SEM.

polymerization in adult *Drosophila* brains results in increased mitophagy, reduced age-associated accumulation of mitochondria, and improved mitochondrial function, as well as cell non-autonomous improvements to muscle mitochondria and proteostasis.

To test if targeting mitochondria function in aging neurons could, reciprocally, affect age-associated brain F-actin polymerization, we examined the impact of knocking down *CG9172*, a complex I subunit gene. Previous work has shown that neuronal RNAi of electron transport chain (ETC) genes can prolong lifespan[68,69]. Here, we show that adult neuronal expression of *CG9172-RNAi* extended *Drosophila* lifespan (Supplementary Fig. 4n) and significantly reduced the number of F-actin-rich rods in brains compared to age-matched controls (Supplementary Fig. 4o, p). These data suggest that mitochondria function may play a role in age-associated brain F-actin polymerization.

Consistent with improved autophagy and mitophagy (Fig. 3g, h; Supplementary Fig. 4a, b), treating middle-aged flies with cytochalasin D significantly reduced the accumulation of mitochondrial content in aged brains (Fig. 4g, h). Notably, less brain mitochondrial content was detected at day 44, after 1 week of cytochalasin D treatment, compared to brains collected from flies of the same cohort at day 37 before treatment (Fig. 4g, h), consistent with a reversal of this hallmark of brain aging. Mitochondrial content was reduced in aged brains at multiple concentrations of the drug (Supplementary Fig. 4q, r). Treating aged flies for 1 week with another actin depolymerization agent, latrunculin A, also resulted in reduced mitochondrial content in aged brains in a dose-dependent manner (Supplementary Fig. 4s, t). Importantly, TMRE staining revealed that depolymerizing age-associated F-actin with cytochalasin D treatment significantly improved mitochondrial homeostasis in the aged brain (Fig. 4i, j). Together, these data indicate that genetic and pharmacological targeting of age-associated F-actin polymerization improves mitophagy and mitochondrial homeostasis in aged *Drosophila* brains.

### Neuronal reduction of F-actin polymerization slows aging via autophagy

Our findings indicate that age-associated F-actin polymerization in the brain disrupts autophagy, mitochondrial homeostasis, and proteostasis. Next, we set out to determine whether the beneficial effects of decreasing actin polymerization on brain and organismal aging are due to improved neuronal autophagy. First, we observed that *Fhos*-mediated modulation of F-actin levels during brain aging proceeds in an autophagy-independent manner. To inhibit the autophagy pathway, we targeted the expression of *Atg1* (Autophagy-related 1, the

*Drosophila* homolog of mammalian ULK1). This Ser/Thr protein kinase regulates the initiation of the formation of the autophagosome[70]. Concomitant knockdown of neuronal *Atg1* and *Fhos* in *elavGS > UAS-Atg1-RNAi,UAS-Fhos-RNAi* flies resulted in reduced age-associated F-actin-rich rods in the brain (Fig. 5a, b). In agreement, pharmacological inhibition of autophagy by chloroquine treatment in *elavGS > UAS-Fhos-RNAi* flies did not block the effect of *Fhos* knockdown in reducing age-associated F-actin-rich rods in the brain (Supplementary Fig. 5a, b). In contrast to extended lifespan with neuronal knockdown of *Fhos* alone (Fig. 2f), induced *elavGS > UAS-Atg1-RNAi,UAS-Fhos-RNAi* flies showed no difference in lifespan compared to vehicle-fed controls (Fig. 5c). Consistent with this finding, we detected no difference in intestinal barrier integrity in flies with neuronal knockdown of both *Fhos* and *Atg1* (Fig. 5d). Hence, although age-associated F-actin polymerization in the brain was reduced with neuronal *Fhos* knockdown, resulting health and lifespan benefits were dependent on functional autophagy.

Furthermore, concomitant knockdown of neuronal *Atg1* and *Fhos* in *elavGS > UAS-Atg1-RNAi,UAS-Fhos-RNAi* flies showed an age-associated accumulation of mitochondrial content in the brain (Fig. 5e, f). Improvements to brain mitochondrial function that were found with disrupting age-associated F-actin polymerization by neuronal *Fhos* knockdown were also lost with concomitant neuronal *Atg1* knockdown (Fig. 5g, h). Together, these results indicate that improved mitochondrial homeostasis associated with reducing F-actin polymerization in aging brains, like improvements to organismal health and lifespan, are dependent on autophagy. Cumulatively, these data are consistent with a model in which age-associated F-actin polymerization in *Drosophila* brains disrupts autophagy and, thereby, drives paradigms of brain and organismal aging.

## Discussion

Actin filaments show a loss of stability and deterioration in aged yeast cells[71] and in multiple non-neuronal tissues of aged *C. elegans*[5]. In contrast, we find a striking age-related increase in F-actin and F-actin-rich rods in the fly brain, which contribute to brain aging and drive organismal health decline. Our findings are consistent with studies showing excess actin stabilization drives neurotoxicity in models of Alzheimer's disease (AD) and related tauopathies[15,16] and Parkinson's disease (PD)[17,72]. As advanced age is a major risk factor for sporadic forms of both AD and PD, actin hyperstabilization may be a shared pathogenic mechanism of age-onset neurodegeneration. Future work will be required to determine the precise cellular mechanisms that lead

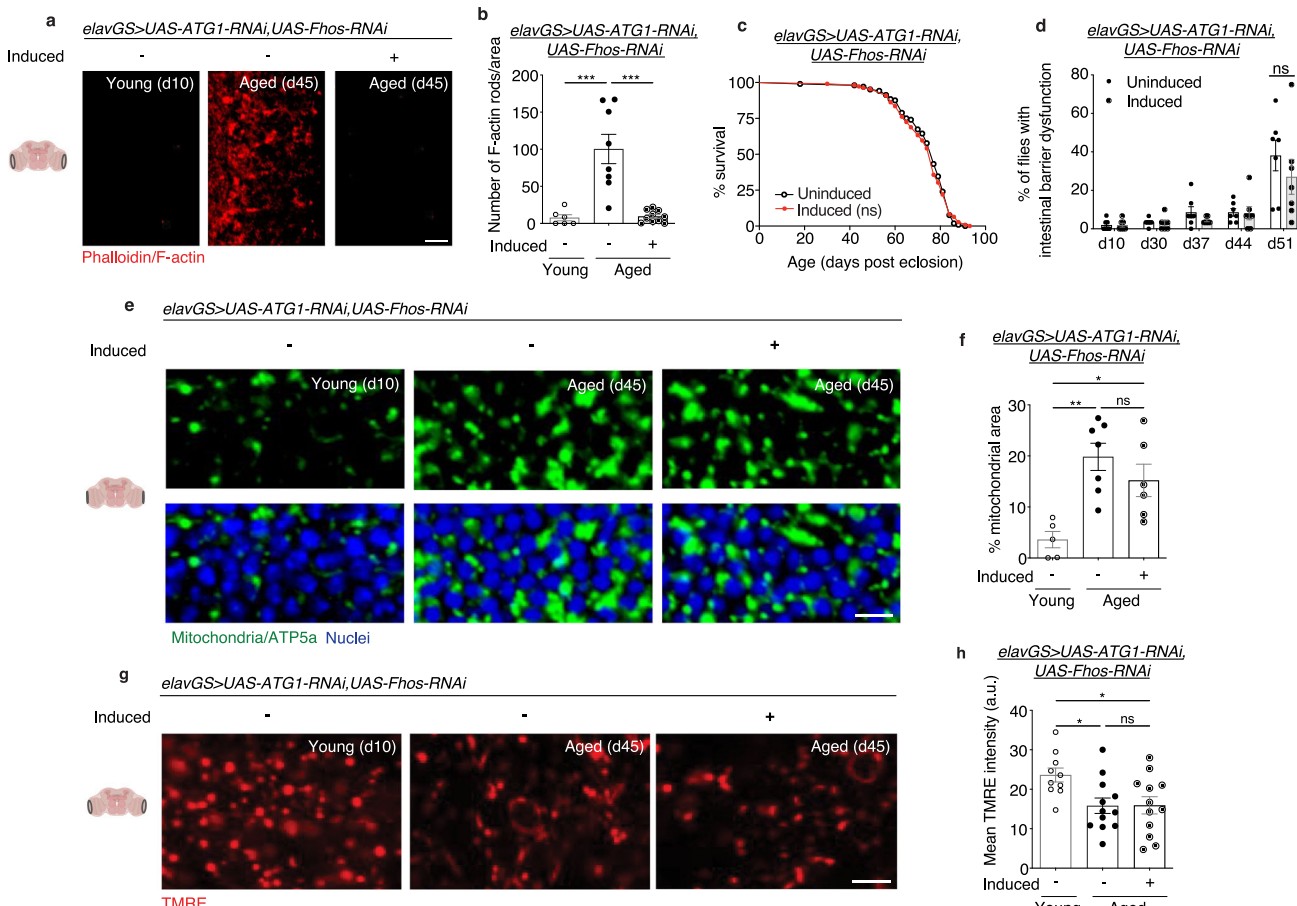

**Fig. 5 | Neuronal reduction of F-actin polymerization slows aging via autophagy. a** Immunostaining of brains at 63x magnification from young (10-day-old) and aged (45-day-old) *elavGS > UAS-Atg1-RNAi,UAS-Fhos-RNAi* flies with or without RU486-mediated transgene expression from day 5 onward, showing F-actin-rich rods (red channel, phalloidin). Scale bar is 5 μm. Accompanying diagrams indicate brain regions where imaging was conducted. **b** Quantification of F-actin-rich rods in brains as shown in **a**. Young *n* = 6, aged uninduced *n* = 8, aged induced *n* = 10 flies, as indicated. ***p* (young vs. aged uninduced) = 0.0001, ***p* (aged uninduced vs. aged induced) < 0.0001, one-way ANOVA/Tukey's multiple comparisons test. **c** Survival curve of *elavGS > UAS-Atg1-RNAi,UAS-Fhos-RNAi* flies with or without RU486-mediated transgene expression from day 5 onward. ns = non-significant, log-rank test. *n* = 180 uninduced and 180 induced biologically independent animals. **d** Intestinal integrity during aging of *elavGS > UAS-Atg1-RNAi,UAS-Fhos-RNAi* flies with or without RU486-mediated transgene expression from day 5 onward. *n* = 7 vials of 30 biologically independent animals per vial on day 10. ns = non-significant; two-way ANOVA/ Šídák's multiple comparisons test. **e** Immunostaining of brains at 63x magnification from young (10-day-old) and aged (45-day-old) *elavGS > UAS-* *Atg1-RNAi,UAS-Fhos-RNAi* flies with or without RU486-mediated transgene induction from day 5 onward, showing mitochondrial morphology (green channel, anti-ATP5a) and nuclear DNA (blue channel, stained with To-Pro-3). Scale bar is 5 μm. **f** Quantification of mitochondrial area in brain as shown in **e**. Young *n* = 5, aged uninduced *n* = 7, aged induced *n* = 6 biologically independent animals, as indicated. **p* = 0.0277, ***p* = 0.0021, ns = non-significant; one-way ANOVA/Tukey's multiple comparisons test. **g** Staining of brains at 63x magnification from young (10-day-old) and aged (45-day-old) *elavGS > UAS-Atg1-RNAi,UAS-Fhos-RNAi* flies with or without RU486-mediated transgene induction from day 5 onward, showing TMRE fluorescence. Scale bar is 5 μm. **h** Quantification of mitochondrial membrane potential measured by TMRE staining as shown in **g**. Young *n* = 10, aged uninduced *n* = 12, aged induced *n* = 12 biologically independent animals, as indicated. *p (young vs. aged uninduced) = 0.0296, *p (young vs. aged induced) = 0.0318, ns = non-significant; one-way ANOVA/Tukey's multiple comparisons test. RU486 or vehicle was provided in the media at a concentration of 50 ug/ml in the indicated treatment groups. Data are presented as scatter plots overlaying mean values +/− SEM.

to excess F-actin and the formation of F-actin-rich rods in the aged brain. Rod-like inclusions (rods) composed of F-actin and actin assembly-regulatory proteins have been shown in cultured neurons to form in response to various stressors, including oxidative stress and mitochondrial energetic impairments[73–75]. As mitochondrial dysfunction is a major hallmark of brain aging, it is reasonable to suggest that there could be a 'vicious cycle' whereby mitochondrial dysfunction and actin hyperstabilization drive brain aging. In addition, it has been shown that de novo actin polymerization is required to form model Hirano bodies[76]. In the context of our study, it is interesting to speculate that actin nucleation is driving excess actin stabilization, and F-actin-rich rod formation, in the aged brain. The most pronounced phenotype that we observe, with respect to lifespan extension, is mediated by neuron-specific RNAi of *Fhos*, which shares the capacity of

other formins to nucleate and bundle actin filaments[40]. We also observe increased expression of actin transcripts in the aged brain. Hence, it is possible that the overall increase in F-actin in the aged brain is due to a combination of increased actin expression and hyperstabilization. Regardless of the mechanisms at play, our findings reveal F-actin accumulation to be an important hallmark of brain aging, which should be considered in the context of existing hallmarks of aging[21].

One of the key hallmarks of brain aging, which drives age-onset pathology is disabled autophagy[21,77] given its demonstrated role in the removal of aggregated proteins and damaged organelles linked to neurodegenerative disease. Hence, identifying interventions that restore autophagy in the aged brain is a promising therapeutic avenue, but one that depends on the step in the process that is impaired and the nature of the impairment. Unfortunately, our understanding of the

nature of autophagy impairments in the aged brain and in most cases of neurodegeneration is limited, making it uncertain whether the same type of intervention is likely to work in all diseases and at all stages of those diseases. Indeed, it has been proposed that dysfunctional autophagy in aged animals, linked to blockage of autophagy at a late stage, may contribute to age-onset health decline[30,78]. Hence, it is possible that interventions that induce early stages of autophagy may not prove effective when targeted to aged animals. In this study, we show that inhibiting F-actin polymerization in aged neurons prevents the age-related loss of autophagic activity in the brain. Moreover, we show that treating middle-aged flies with an actin polymerization inhibitor, cytochalasin D, restores brain autophagy to pre-treatment levels and leads to lower amounts of protein aggregates and mitochondrial content in the brain than before the treatment began. That said, we failed to observe a longevity phenotype in response to cytochalasin D treatment, possibly due to detrimental effects in other organ systems. Interestingly, our data indicate that the accumulation of F-actin in the aged brain is not due to impaired autophagy. We show that the ability of neuronal *Fhos* inhibition to prevent F-actin accumulation in the aged brain is autophagy-independent. However, reducing F-actin levels in the aged brain while disrupting autophagy fails to improve mitochondrial homeostasis in the aged brain or prolong healthspan. Hence, our current working hypothesis is that disrupting actin polymerization in aging neurons prolongs healthspan via improvements in brain autophagy. It is also worth considering that the age-related change in actin structure could lead to a remodeling of neuronal structure (e.g., axonal remodeling).

The functional significance of abnormal F-actin accumulation in cellular health and disease is highlighted by clinical data studying neurodegenerative diseases. There is an emerging consensus that F-actin-containing intracellular inclusions disrupt neuronal function and are a likely a cause of synaptic loss without neuronal loss, as occurs early in dementias[79,80]. Here, we show that disrupting actin polymerization in the brains of middle-aged animals robustly improves a well-established paradigm of olfactory learning[51,81]: the ability of flies to associate an odor with an aversive stimulus. Our findings reveal that inhibiting actin polymerization in aged animals can slow or even reverse aspects of brain aging. It is worth noting that previous work has shown that spermidine feeding can suppress age-onset memory impairment in an autophagy-dependent manner[82]. It is important to consider, however, that it is unlikely that disrupting actin polymerization in every cell and tissue type would promote organismal health. It is likely that actin polymerization is essential to cell and tissue homeostasis in numerous contexts. Hence, in order to translate these findings to benefit human health, future work could focus upon identifying cell-type and tissue-specific approaches to target actin polymerization in aged organisms.

## Methods

### Fly stocks
The fly strain *Elav−GeneSwitch* (*ElavGS*) was provided by H. Keshishian (Yale University, New Haven, CT, USA). GFP-mCherry-Atg8a was provided by Eric Baehrecke (University of Massachusetts Medical School, Worcester, MA, USA). *UAS-mito-QC* was provided by Alexander J. Whitworth (University of Cambridge, UK). *tub-Gal80ts;elav-Gal4* was provided by Julie Secombe (Albert Einstein College of Medicine, NY, USA). *NP6293-Gal4* (PG) was provided by Marc Freeman (Vollum Institute, Oregon Health & Science University, USA). *UAS-Fhos-RNAi* (31400, 51391), *UAS-tsr* (20665), *UAS-Act5c-GFP* (7309), *UAS-Act42a-GFP* (9252), *UAS-dsRNA-GFP* (9330), *Mdr65-GAL4* (50472, SPG), *GMR54H02-Gal4* (45784, CG), *alrm-Gal4* (67031, ALG), *rumpel-Gal4* (77469, EG), and *Atpα-GeneSwitch* (59948, GlialGS) were acquired from the Bloomington *Drosophila* Stock Center. *UAS-Act5c-RNAi* (101438), *UAS-Act42a-RNAi* (104731), and *UAS-Atg1-RNAi* (16133) lines were received from Vienna *Drosophila* Resource Center (VDRC).

### Fly Husbandry and Lifespan Analysis
Flies were maintained in vials containing cornmeal medium (1% agar, 3% yeast, 1.9% sucrose, 3.8% dextrose, 9.1% cornmeal, 1.1% acid mix, and 1.5% methylparaben, all concentrations given in wt/vol). Flies were collected under light anesthesia by nitrogen gas and housed at a starting density of 30 fertilized female flies per vial. All flies were kept in a humidified, temperature-controlled incubator with a 12 h:12 h dark:light cycle at 25 °C. For experiments using the *tub-Gal80ts;elav-Gal4* line, flies were crossed at 18 °C and developing embryos and larva were maintained at 18 °C until 3 days post eclosion of adult stage, when they were moved to 29 °C. RU486 was dissolved in ethanol and administered in the media as indicated while preparing food. Cytochalasin D (Sigma-Aldrich C2618), latrunculin A (Sigma-Aldrich 428026), chloroquine (InvivoGen tlrl-chq-4), jasplakinolide (Sigma-Aldrich 420127), DMSO vehicle, or water vehicle were mixed in media as indicated. Flies were flipped to fresh vials every 2–3 days and scored for death.

### Immunostaining and image analysis
For brain and muscle immunostaining, flies were fixed in 3.7% formaldehyde in phosphate-buffered saline (PBS) for 20 min. After fixation, brains and hemi-thoraces were dissected and fixed again for 5 min. Samples were then rinsed 3 times for 10 min with PBST and blocked in 3% BSA in PBST (PBST-BSA) for 1 hour. Primary antibodies were diluted in PBST-BSA and incubated with samples overnight at 4 °C. Primary antibodies used were: mouse-anti-ATP5a 1:250 (15H4C4, Abcam); mouse anti-actin 1:50 (JLA20, DSHB), mouse-anti-FK2 1:250 (BML-PW8810-0500, ENZO); rabbit-anti-atg8a 1:250 (Rana et al., 2017), rabbit-anti-ref(2)P 1:250 (p62, Rana et al., 2017), mouse-anti-dsDNA 1:250 (ab27156, Abcam), rabbit-anti-DILP2 1:250 (a generous gift from Dr. Seung Kim), mouse-anti-neuroglian 1:50 (BP 104, DSHB), and rat-anti-N-cadherin 1:50 (DN-Ex #8, DSHB). Samples were then rinsed 3 times in PBST for 10 min and incubated with the secondary antibodies and/or stained at room temperature for 3 hours. Secondary antibodies and stains used were: anti-rabbit or anti-mouse AlexaFluor-488 1:500 (A-11001 or A-11008, Thermo Fisher Scientific); anti-rabbit, anti-mouse, or anti-rat AlexaFluor-568 1:500 (A-11031, A-11036, or A-11077, Thermo Fisher Scientific); anti-mouse AlexaFluor-647 1:500 (A-21235, Thermo Fisher Scientific); To-Pro-3 DNA 1:500 (T3605, Thermo Fisher Scientific); DAPI (diamidino-2-phenylindole) 1:1000 (62247, Thermo Fisher Scientific); phalloidin AlexaFluor-568 1:200 (A12380, Thermo Fisher Scientific); and phalloidin AlexaFluor-488 1:200 (A12379, Thermo Fisher Scientific). Finally, samples were rinsed 3 times with PBST for 10 min and mounted in Vectashield Mounting Medium (Vector Lab). Images were taken using a Zeiss LSM780 or LSM880 confocal microscope and analyzed with ImageJ software to measure intensity, area, count, and/or sizes of stained structures. Diagrams of *Drosophila* brains accompanying microscopy images were created with BioRender.com to show where the imaging took place.

### TMRE staining
Flies were anesthetized and dissected in cold *Drosophila* Schneider's Medium (DSM). Brains and hemi-thoraces were incubated in TMRE staining solution (100 nM TMRE (T669, Thermo Fisher Scientific) in DSM) for 12 min at room temperature. After staining, samples were rinsed once in wash solution (25 nM TMRE in DSM) for 30 seconds before being mounted in wash solution. Images were acquired using a Zeiss LSM880 confocal microscope and TMRE intensity was quantified using ImageJ software.

### GFP-mCherry-Atg8a tandem and Mito-QC
Flies were anesthetized and dissected in cold Drosophila Schneider's Medium (DSM). Brains were mounted in DSM solution. Images were acquired on a Zeiss LSM780 or LSM880 confocal microscope and

autolysosomes or mitolysosomes (mCherry-only foci) were quantified using ImageJ software.

## Olfactory training

Aversion training was performed as described in[51] using a system from MazeEngineers (Conduct Science). Briefly, flies were exposed to a neutral odor (3-octanol) by air pump in a training chamber for one minute under low red light conditions. They were then exposed to the odor in a series of twelve 60-V shocks for 1.25 seconds followed by rest for 3.75 seconds for a total of one minute. Flies recovered for one hour before being placed in a T-maze with the trained scent on one side and a second neutral scent (4-methylcyclohexanol) on the other side of the maze. After two minutes of exploration under dim red light conditions, flies in either chamber of the maze were counted.

## Intestinal barrier dysfunction (Smurf) assay

Intestinal integrity assays were performed as previously described[57]. Flies were aged to the indicated time points with standard RU- or RU+ food as indicated. To conduct the "Smurf" assay, flies were then transferred to new vials containing standard medium with 2.5% wt/vol F&D blue dye # 1 (SPS Alfachem) for 16 hours. The number of flies per vial with dye coloration outside the gut (Smurf flies) were then tallied and quantified.

## Climbing activity assays

In negative geotaxis assays, flies were gently tapped to the bottom of 10 cm vials. After 10 seconds, the number of flies that climbed above 5 cm were recorded. For forced climbing assays, 100 adult flies from each treatment group were placed in 200 ml glass cylinders. The cylinders were tapped quickly and the flies were allowed to settle for 2 minutes. This step was repeated nine times. 1 minute after the final tap, the number of flies in the upper, middle, and lower third of the cylinder was recorded.

## Spontaneous physical activity assay

Vials containing 10 adult flies were placed inside a *Drosophila* activity monitor (TriKinetics). Movements were recorded continuously under normal culturing conditions for 36 hours on a 12 h:12 h dark:light cycle. Graphs represent mean activity per fly per hour and the scatterplot shows spontaneous activity per fly during a 12 h:12 h dark:light cycle. Triplicate samples were used for each activity measurement.

## Consumption-excretion (Con-Ex) assay

Con-EX assays were performed as previously described[52]. Adult flies were transferred to new empty vials (10 flies per vial with a total of 6 vials) and fed from feeder caps containing standard medium with 2.5% wt/vol F&D blue dye # 1 for 20 h at 25 °C. Feeder caps were discarded at the conclusion of feeding. For checking internal (consumed) dye, flies were homogenized in 500 ul of ddH$_2$O and pellet debris were removed by centrifugation. The dye excreted by the flies on the walls of the vials was collected by adding 1 ml of ddH2O to each vial and vortexing. Samples were quantified using an Epoch BioTek microplate reader and compared to a serially diluted standard.

## Quantitative real-time PCR

Total RNA was extracted from samples using TRIzol reagent (Invitrogen) following manufacturer protocols. Samples were treated with DNAse before cDNA synthesis was performed using the First Strand cDNA Synthesis Kit from Fermentas. qPCR was performed using Power SYBR Green master mix (Applied Biosystems) on a BioRad Real-Time PCR system. Cycling conditions were as follows: 95 °C for 10 min; 95 °C for 15 s then 60 °C for 60 s, cycled 40 times. GAPDH or RPL32, as indicated, were used as reference genes to normalize. BioRad CFX Manager ver. 3.1 was used to collect and analyze qRT-PCR data. Primer sequences used are as follows:

GAPDH, GCGGTAGAATGGGGTGAGAC and TGAAGAGCGAAAACAGTAGC

RPL32, GACCATCCGCCCAGCATAC and CGGCGACGCACTCTGTT

Act5c, AGGCCAACCGTGAGAAGATG and GGGGAAGGGCATAACCCTC

Act42a, ATGGTAGGAATGGGACAAAAGGA and CTCAGTAAGCAAGACGGGGTG

4E-BP, TACACGTCCAGCGGAAAGTT and CCTCCAGGAGTGGTGGAGTA

## Enzyme-linked immunosorbent assays (ELISAs)

ELISA specific for F-actin (MyBiosource) was used according to manufacturer protocol. Five fly heads were homogenized in 30 μl of homogenizing F-actin stabilization buffer (Cytoskeleton). The starting concentration of total protein isolates from heads was assessed by Bradford assay (Thermo Fisher Scientific) according to manufacturer protocol. Samples were diluted in sample diluent at a ratio of 1:20 before being loaded to a microplate that was pre-coated with an antibody specific for F-actin. A biotinylated secondary antibody was added to the microplate and subsequently incubated with HRP-avidin followed by peroxidase substrate. Concentrations of protein and F-actin were determined using an Epoch BioTek microplate reader and compared to a serially diluted standard provided by the manufacturer.

## Statistics

GraphPad Prism 10 (GraphPad Software, La Jolla, CA, USA) was used to perform statistical analysis and graphically display data. Significance is expressed as $p$ values as determined by two-tailed, unpaired, parametric, or non-parametric tests as indicated in figure legends. When comparing two groups, unpaired t-tests were used when data met criteria for parametric analysis and Mann-Whitney tests were used for non-parametric analysis. To compare more than two groups when parametric tests were appropriate, one-way ANOVAs with Tukey's multiple comparisons tests were performed. To compare more than two groups sampled from a Gaussian distribution without assuming equal variances, Welch and Brown-Forsythe ANOVAs were used. To analyze more than two groups when data did not meet requirements for parametric tests, Kruskal-Wallis tests with Dunn's multiple comparisons post hoc tests were used. When performing grouped analyses with multiple comparisons, two-way ANOVAs with Šídák's multiple comparisons test were performed. Bar graphs depict mean ± standard error of the mean (SEM). The number (n) of biological samples used in each experiment can be found in figures and figure legends. Log-rank (Mantel-Cox) tests were used to compare survival curves. No statistical methods were used to pre-determine sample sizes but our sample sizes are similar to those reported in previous publications[60,64]. Blinding was performed when possible, specifically when conducting microscopy for TMRE, Atg8a-tandem, and mitoQC. Blinding was not always possible during experimental setup due to the need to carefully document the genotypes of flies when generating crosses or to track groups assigned RU486/drugs vs. vehicle throughout lifespans. All experiments were conducted under the same conditions, and control and experimental samples were treated equally and in parallel to exclude bias. Additionally, all images were taken in the same location and depth in each tissue type. Parents of experimental flies were randomly grouped into mating vials with 10 virgin females to 7 mature males. Upon eclosion, experimental flies were randomly assigned to mating bottles (10 vials per bottle) for 3 days. These bottles were then sorted into vials containing 30 mated females each before evenly distributing these vials assigned randomly into treatment and control groups. No animals or data points were excluded from the analyses. The difference between two groups was defined as statistically significant for the following $p$ values: *$p < 0.05$, **$p < 0.01$, ***$p < 0.001$ (and non-significant when $p > 0.05$).

**Reporting summary**

Further information on research design is available in the Nature Portfolio Reporting Summary linked to this article.

## Data availability

All data generated or analyzed during this study are included in the figures and text, with representative images accompanying quantified results where applicable, unless otherwise noted. Further information is available from the corresponding author upon reasonable request. Source data are provided with this paper.

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

## Acknowledgements

We thank E. Baehrecke (UMass Medical School), Marc Freeman (Vollum Institute), Alex Whitworth (Cambridge University), the Vienna *Drosophila* Resource Center and the Bloomington Drosophila Stock Center (NIH no. P40OD018537) for fly stocks; and R. Aparicio for technical assistance. We thank N. Prunet and the MCDB/BSCRC Microscopy Core for training and microscope facilities. This work was supported by NIH grants R01AG037514 and R01AG049157 to D.W.W.

## Author contributions

E.T.S. and D.W.W. designed the experiments. J.M.S. conducted rapamycin experiments and actin transcript analysis. N.L-A. helped conduct muscle TMRE experiments, *4E-BP* transcript analysis, and Bradford

assays. K.S.W. helped conduct latrunculin A drug experiments. E.T.S. conducted all other experiments and analyzed the data. E.T.S. and D.W.W. wrote the manuscript.

## Competing interests

The authors declare no competing interests.
