## [Peer Review File · Nature Communications]

REVIEWER COMMENTS

Reviewer #1 (Remarks to the Author):

This manuscript from David Walker's research group describes the exciting observation that excess actin polymerization is a novel biomarker for brain aging. Specifically, they show that brain aging in *Drosophila* is associated with an accumulation of filamentous actin (F-actin) and that this can be suppressed by pro-longevity cues. They further define the crucial role autophagy plays in this process. A major strength of the study is the fact that it is the first to link actin dynamics to brain aging. Furthermore, the multiple genetic tools and reagents used would be directly applicable to studying actin dynamics in other organs such as the muscle and gut. For instance, Act88F regulates indirect flight muscle formation and may have a muscle-specific role; the relevance of this gene to muscle aging can now be explored using techniques analogous to what is described in this paper. Similarly, the relevance of other actin isoforms such as Act57B, Act79B, Act87E and Activin-beta to aging can also now be explored. In addition, different isoforms of the actin cross-linking protein, alpha-actinin, are expressed in the muscle, ring canal of the ovary, etc. – the significance of these differences to actin dynamics and aging of the relevant tissues can now be investigated. Actin also has many posttranslational modifications, including acetylation, phosphorylation, arginylation, glutathionylation, etc. – which can now be explored for possible roles in *Drosophila*. Thus, this manuscript is highly significant as it describes a paradigm-shifting study that will provide an opportunity to answer many questions about aging, healthspan and actin dynamics in multiple *Drosophila* tissues. However, while the methodology described is adequate and most conclusions drawn are sound, I have a few concerns that I would like to see addressed prior to publication of this outstanding body of work.

Major

1. Davis Walker's research group previously elegantly described a series of mitochondrial OXPHOS disruptions in the nervous system that could increase lifespan (Copeland et al., 2009). Can the authors show whether any of the specific scenarios of enhanced longevity in the Copeland et al. manuscript is associated with a clearance of age-associated actin-rich rods?

2. Figure 2 describes a very important finding in this manuscript, and I would like to see it further strengthened. Is the age-dependent accumulation of poly-ubiquitin aggregates in muscles ameliorated in *ElavGS>UAS-Fhos-RNAi* flies? The authors should also compare transmission electron micrographs of thoraces of aged *ElavGS>UAS-Fhos-RNAi* flies +/- RU486 to get a sense of mitochondrial integrity.

3. What is the effect of Fhos KD on systemic insulin signaling? Does it impact the release of Dilp2 from the brain? How does it affect markers of insulin signaling in the brain, gut, and thorax? Use immunoblotting to phospho-AKT and qPCR of a gene known to be robustly induced in response to reduced insulin signaling, such as thor (4E-BP) in the brain, gut and thorax to address this question.

4. It is important to test whether knocking down Fhos or Actin 5C in the nervous system using Elav-Gal4 and the tub-Gal80-ts system (raised at 18C during development and 29C as adults) will reproduce the extended lifespan data? If no extended lifespan is seen, then perhaps the extended longevity phenotype is not reproduced due to thermal stress at 29C. This should be stated in the Discussion section.

5. Is the extended lifespan reproduced by Cytochalasin D and/or Latrunculin A? If not, please explain possible reasons why in the Discussion section.

6. To corroborate the results in Figure 5A, please examine the effect of a chemical inhibitor of autophagy, such as chloroquine.

Minor

Line 29, Summary – the word modulating (which can mean increasing or decreasing) should be changed to “decreasing” to eliminate any ambiguity.

Line 218-220 should be re-phrased as *tsr* was overexpressed (not knocked down).

Line 231 should be re-phrased; the result described is indicative of reduced autophagic flux (instead of activity), as it may be confusing to the non-specialist that an increase in autophagosomes as clearly shown in the image is indicative of reduced autophagic activity.

Line 271 – (Fig i,j) should be (Fig 3i,j).

Line 651 – the period after 10 mins should be deleted.

Lines 654-655 – the text needs to be rectified.

Figure 5E – figure label, ATG1-RNAI should be ATG1-RNAi.

Why was canton S used in some panels and Dahomey in others? Is there any reason why Oregon R flies were not tested?

Reviewer #2 (Remarks to the Author):

The authors describe that based on phalloidin stainings, filamentous actin (F-actin) increases in the aging *Drosophila* brain (from day 10 to day 30). These accumulations were reversed with »longevity interventions«, dietary restriction (d21 comparison), and rapamycin treatment (10 d vs 45 d). To genetically interfere with F-actin accumulation, they after having done a miniscreen decide on neuronally knocking down (KD) a specific Formin (*elavGS>UAS-Fhos-RNAi*), which at d=45 reduces the brain phalloidin staining intensity. In parallel, they also use cytochalasin »mid-age« feeding to reduce phalloidin label. They then document »benefits« of the Fhos KD in terms of lifetime but also using olfactory conditioning memory testing (1 hr aversive memory) as well as assays of gut barrier integrity and locomotor activity. They then monitor autophagy in their context, using ATG8 immunostainings as well as a tandem GFP-mCherry-ATG8a sensor. They interpret their findings as loss of F-actin restoring brain/neuron autophagy. Next, they address mitochondrial status and the mitophagic situation, documenting an increase of »mito-lysosomes« and a decrease of ATP5a upon pan-neuronal Fhos KD. Finally, they combine KD of ATG1 with their Fhos KD. While the reversal of the age induced F-actin rod accumulation is still present here, neither the longevity improvement nor the decrease of ATP5A label in aged animals after Fhos KD are anymore observed.

Overall: I do find their most important finding of reversing the F-actin accumulation in aging *Drosophila* (brains) to be »protective« very interesting, so that the manuscript in principle could be a candidate for Nature Comm. This said, I am somewhat less convinced by their data stating that a restauration of brain autophagy/mitophagy is necessarily responsible here. This also as any mechanistic suggestion of how this crosstalk (F-actin with (mito)autophagy) would operate is missing. In this regard, at least some effort concerning the sub-cellular nature of the F-actin accumulation would have been appropriate. Moreover, it do have some concerns with some of the autophagy/mitophagy-related data and their interpretation.

Points:

- Concerning the autophagy assays: is an accumulation of ATG8 puncta really accepted to be an indication of their being autophagy defects? Related: given that they find more ATG8 puncta in the anti-ATG8 immunofluorescence staining (Fig. 4A), shouldnt this also speak up when using the tandem ATG8 GFP/mCherry sensor, as the sum/total integral of mCherry/GFP signals being increased in aged brains? I did find making the »red-only area« the only parameter shown a

somewhat selective. How about the red/green ratios? It is a game of emphasizing total versus relative amounts of autophagic vacuole (maturation), but a normalization of mCherry-only spots to total mCherry-GFP-signal would be informative here.

- For the »mito-lysosomes« (Fig. 4A): isn't there also more GFP signals in the aged Fhlos KD, at least their images suggest this? How about the ratio GFP/Cherry ratio here? From what I do see I am afraid I cannot agree on there being more (means a higher ratio) mitochondria in lysosomes (side note, knowing that previous papers indeed : the term »mitolysosome« in my ears is somewhat confusing as it implies lysosomes within/at mitochondria not vice versa, though I do know it was used in the literature in some papers like that). Fig. 4A: why are no young control animals shown?

- Their cellular/cellbiological description of the F-actin rods (definition of rods?) is very shallow/essentially missing. Is there an accumulation in neuronal cell bodies, axons and dendrites (or at least in the »neuropile«) to the same extent? Are their rods displaced as compared to the physiological localization of F-actin? And: where in the brain are their images (Fig. 2c ...) taken from? Always the same ?

- Instead of using here »young/old« and at other places specifying the exacte age (e.g. 21 d), I suggest they should always give the exacte age of the flies, as the exacte age is relevant for interpretation. It looks as if the choice of analysis time points was somewhat arbitray (Fig. 1g,h,i vs Fig. J,k,l)

- They did ELISA from head homogenates of young vs old flies and present an actin-F accumulation (from what I do see there was no normalization to protein amounts, just to fly head number, correct? Would be interesting to see total protein normalized data; also: analyzing dissected brains instead of heads would be more appropriate in my eyes). Same time, they observe an upregulation of actin mRNA. Isn't that a likely source of the increased F-actin label? Is the transcriptional increase a rather special feature, or is this found for a majority of analyzed transcripts in an RNAseq experiment? How about testing heterozygosity of an actin locus to suppress aspects of aging?

- There is a body of Drosophila literature concerning the age-induced decline of olfactory memory scores, the role of autophagy herein, and the protective effects of dietary restriction mimetics. I think it would be appropriate to cite at least some of this work. Along the same lines: what they apparently do measure for my understanding is one hour aversive mid-term olfactory memory, I think they should stay with normally used nomenclature here, again they are somewhat disregarding previous work. Per se, there would be the possibility to separate anesthesia sensitive from resistant by post.-training amnesic cooling, means to separate ASM from ARM. So far, it was ASM found to be age-sensitive. They thus could better connect their data to existing literature.

- Fig.2g vs 5g: why is the absolute lifetime of the two controls that different? Looks like 50 (2g) vs 75 (5g) days LD50. The elavGS>UAS-ATG1-RNAi,UAS-Fhos-RNAi seem to have the same lifetime as the elavGS> UAS-Fhos-RNAi. I am prepared to accept that absolute lifetimes measured are often somewhat variant, however, this difference between controls worries me that the effect is not necessarily due to the KD of ATG1 here...

- There were many labeling mistakes etc..care should ve taken in case of a revision..

- There is a previous paper (Anhezini, L., Saita, A.P., Costa, M.S., Ramos, R.G., Simon, C.R. (2012). Fhos encodes a Drosophila Formin-Like Protein participating in autophagic programmed cell death. *genesis* 50(9): 672—684), speaking about the nuclear transfer of Formin (and more literature reporting nuclear functions of Formins). How sure can they be that their formin KD longevity phenotype really is due to the loss of F-actin (I do see that they used a pharmacological assay as well, however, they report less age-protection findings here)? How about the results of their actin modulator mini-screen..any more supportive evidence from this approach?

- Spelling mistakes such as 72. 'mitochondrial', 86. 'including by', 136. 'brains'

Reviewer #3 (Remarks to the Author):

In this manuscript, Schmid and colleagues examined the role of actin dynamics in the aging brain using *Drosophila* model. They find a great increase of filamentous actin in aging fly brains. Through genetic and pharmacological manipulation of actin polymerization, the authors conclude that excess actin polymerization disrupts neuronal homeostasis (such as autophagy, mitophagy, mitochondrial function), while neuronal knockdown of polymerization factor Fhos extends lifespan and preserves neuronal function with age. Thus, the work uncovers a novel interplay between actin polymerization and brain aging. I have a few concerns as listed below.

1. The lifespan extending effect of neuronal knockdown of Fhos is a very interesting finding. The authors mentioned that they also tested several other actin polymerization factors. I suggest the authors include these results in the paper. It is important to demonstrate whether the lifespan is regulated by specific (not general) actin polymerization factors.

2. There is no justification for why the pan neuronal driver is used in the genetic manipulations of polymerization factors. The authors show that actin increases in the aging brain. However, it is unclear whether this increase mainly occurs in neurons or other cell types of the central brain, such as glial cells. The authors need to demonstrate whether the elevated actin polymerization is found in aging neuronal tissue, and/or glial cells. It would be great if they could also test if neuronal knockdown of Fhos attenuates age-associated actin rods in neuronal tissues or glial cells using cell type-specific markers.

It is clear that actin contents increase in the aging brain. But how exactly aging alters actin contents was not fully addressed. Even though the results from Fhos knockdown experiments suggest a possible involvement of actin polymerization, the age-related change of actin structure is so dramatic and it is highly likely the remodeling of neuronal structure (e.g., axonal remodeling) is another underlying cause. The authors should discuss this possibility in the manuscript.

3. It appears that most of the imaging analysis was focused on the optic lobe, in particular the medulla layer. I wonder if the changes of age-associated actin rods are also prominent in other regions of the brain, for example, antennal lobe and mushroom bodies. The reason I am asking this is that the authors did a learning and memory test using olfaction aversion training, which is associated with the function of the antennal lobe and mushroom bodies. Can the authors check if antennal lobe and mushroom bodies also exhibit age-associated increases in actin rods, and if neuronal knockdown of Fhos could attenuate it?

As a side note, the authors should clarify in the text or figure legend which regions of the brain were used in each fluorescence image, including all actin staining, autophagy, mitophagy, and mitochondrial images.

4. It is a little bit counterintuitive that inhibition of actin polymerization can reverse age-related decline in autophagic flux. As mentioned by the authors, actin dynamics are essential in the biogenesis and transportation of most cellular vesicles, including autophagosomes. To further demonstrate that excess actin polymerization is detrimental and impairs autophagic activity, the authors could genetically enhance actin polymerization or treat flies with actin polymerizing and stabilizing drugs and then test if excess actin polymerization blocks autophagy and mitophagy in the brain.

5. The authors used an anti-dsDNA antibody to visualize mtDNA. How specific is this antibody? Have the authors verified its specificity experimentally (to exclude the cross-reaction with nuclear dsDNA)?

6. Minor issues:

Fig 4a,b young age data are needed. Why is the GFP-positive structure ring shape, but the mCherry structure exhibits puncta shape?

Line 293, "Extended Data Fig 3c,d" should be "Extended Data Fig 4c,d"

Line 295, "Extended Data Fig e,f" missing figure number

We thank the reviewers for their constructive comments and suggestions. In response, we have edited the manuscript substantially, including the addition of a very large amount of new data. We sincerely thank the reviewers for helping to improve our manuscript.

Reviewers' Comments:

Reviewer #1 (Remarks to the Author):

This manuscript from David Walker's research group describes the exciting observation that excess actin polymerization is a novel biomarker for brain aging. Specifically, they show that brain aging in *Drosophila* is associated with an accumulation of filamentous actin (F-actin) and that this can be suppressed by pro-longevity cues. They further define the crucial role autophagy plays in this process. A major strength of the study is the fact that it is the first to link actin dynamics to brain aging. Furthermore, the multiple genetic tools and reagents used would be directly applicable to studying actin dynamics in other organs such as the muscle and gut. For instance, Act88F regulates indirect flight muscle formation and may have a muscle-specific role; the relevance of this gene to muscle aging can now be explored using techniques analogous to what is described in this paper. Similarly, the relevance of other actin isoforms such as Act57B, Act79B, Act87E and Activin-beta to aging can also now be explored. In addition, different isoforms of the actin cross-linking protein, alpha-actinin, are expressed in the muscle, ring canal of the ovary, etc. – the significance of these differences to actin dynamics and aging of the relevant tissues can now be investigated. Actin also has many posttranslational modifications, including acetylation, phosphorylation, arginylation, glutathionylation, etc. – which can now be explored for possible roles in *Drosophila*. Thus, this manuscript is highly significant as it describes a paradigm-shifting study that will provide an opportunity to answer many questions about aging, healthspan and actin dynamics in multiple *Drosophila* tissues.

Thanks for recognizing the novelty, significance and “paradigm-shifting” nature of our study. We sincerely appreciate your careful and thorough evaluation of our manuscript.

However, while the methodology described is adequate and most conclusions drawn are sound, I have a few concerns that I would like to see addressed prior to publication of this outstanding body of work.

Major

1. Davis Walker's research group previously elegantly described a series of mitochondrial OXPHOS disruptions in the nervous system that could increase lifespan (Copeland et al., 2009). Can the authors show whether any of the specific scenarios of enhanced longevity in the Copeland et al. manuscript is associated with a clearance of age-associated actin-rich rods?

Thanks for this interesting suggestion. We have now examined whether ETC-mediated longevity (using one of the interventions from Copeland et al, 2009) impacts actin dynamics in the aged brain. As shown in Supplementary Fig. 4n,o,p, we show that this mode of lifespan extension is also linked to reduced actin-rich rods in the aged brain!

2. Figure 2 describes a very important finding in this manuscript, and I would like to see it further strengthened. Is the age-dependent accumulation of poly-ubiquitin aggregates in muscles ameliorated in *ElavGS>UAS-Fhos-RNAi* flies? The authors should also compare transmission electron micrographs of thoraces of aged *ElavGS>UAS-Fhos-RNAi* flies +/- RU486 to get a sense of mitochondrial integrity.

Thanks for this interesting suggestion. We have now examined poly-ubiquitin aggregates in muscles of flies with neuronal Fhos RNAi. Interestingly, in Supplementary Fig. 4e,f,g, we show that *ElavGS>UAS-Fhos-RNAi* induction improves proteostasis in aging muscles! We have gone beyond the suggestion to look at mitochondrial integrity via EM and have additionally examined mitochondrial function and morphology via immunofluorescence microscopy. As shown in Supplementary Fig. 4h,i, we show that this neuronal intervention improves mitochondrial homeostasis in aging muscles!

3. What is the effect of Fhos KD on systemic insulin signaling? Does it impact the release of Dilp2 from the brain? How does it affect markers of insulin signaling in the brain, gut, and thorax? Use immunoblotting to phospho-AKT and qPCR of a gene known to be robustly induced in response to reduced insulin signaling, such as *thor* (4E-BP) in the brain, gut and thorax to address this question.

This is an intriguing question. We have now examined DILP2 levels in insulin producing cells (IPCs) in the brains of Fhos KD flies and controls. As shown in Supplementary Fig. 2e,f, we did not observe any significant difference between DILP2 levels in IPCs of control and Fhos KD flies. Furthermore, we didn't observe any difference in FOXO activation/4E-BP mRNA levels systemically (gut, thorax) or in the brain (Supplementary Fig. 4g,h,i). Hence, we conclude there is not a major alteration with regard to insulin signaling in response to neuronal Fhos KD.

4. It is important to test whether knocking down Fhos or Actin 5C in the nervous system using *Elav-Gal4* and the *tub-Gal80-ts* system (raised at 18C during development and 29C as adults) will reproduce the extended lifespan data? If no extended lifespan is seen, then perhaps the extended longevity phenotype is not reproduced due to thermal stress at 29C. This should be stated in the Discussion section.

We have now carried out this experiment. As shown in Supplementary Fig 4j, we find a very striking increase in longevity upon adult-onset Fhos RNAi using Elav-Gal4 and the GAL80-ts system! As noted, this strengthens our conclusions. Thanks for this suggestion.

5. Is the extended lifespan reproduced by Cytochalasin D and/or Latrunculin A? If not, please explain possible reasons why in the Discussion section.

We agree that this is an intriguing question. So far, we have yet to find a Cytochalasin D or Latrunculin A treatment that can extend lifespan. As you can imagine, we presume this to be due to the effects of disrupting actin dynamics throughout the entire body having detrimental effects. We have added to the Discussion section in this regard.

6. To corroborate the results in Figure 5A, please examine the effect of a chemical inhibitor of autophagy, such as chloroquine.

We have now done so. In Supplementary Fig. 5a,b, we show that chloroquine doesn't prevent Fhos KD-mediated alterations in actin dynamics. This is, of course, in line with our genetic (RNAi-based) findings. Thanks for this suggestion which strengthens our conclusions.

Minor

Line 29, Summary – the word modulating (which can mean increasing or decreasing) should be changed to “decreasing” to eliminate any ambiguity.

Done

Line 218-220 should be re-phrased as *tsr* was overexpressed (not knocked down).

Done

Line 231 should be re-phrased; the result described is indicative of reduced autophagic flux (instead of activity), as it may be confusing to the non-specialist that an increase in autophagosomes as clearly shown in the image is indicative of reduced autophagic activity.

Done

Line 271 – (Fig i,j) should be (Fig 3i,j).

Done

Line 651 – the period after 10 mins should be deleted.

Done

Lines 654-655 – the text needs to be rectified.

Done

Figure 5E – figure label, ATG1-RNAI should be ATG1-RNAi.

Done

Why was canton S used in some panels and Dahomey in others?

We wanted to examine actin dynamics during aging in more than one lab strain to determine the generality of the findings. Dahomey was used to validate our findings with the initial lab strain (Canton S) that we examined. In addition, we have examined actin dynamics in many additional genetic backgrounds by utilizing multiple transgenes, etc. For no specific reason, we don't work with Oregon R in the lab

Reviewer #2 (Remarks to the Author):

The authors describe that based on phalloidin stainings, filamentous actin (F-actin) increases in the aging *Drosophila* brain (from day 10 to day 30). These accumulations were reversed with »longevity interventions«, dietary restriction (d21 comparison), and rapamycin treatment (10 d vs 45 d). To genetically interfere with F-actin accumulation, they after having done a miniscreen decide on neuronally knocking down (KD) a specific Formin (*elavGS>UAS-Fhos-RNAi*), which at d=45 reduces the brain phalloidin staining intensity. In parallel, they also use cytochalasin »mid-age« feeding to reduce phalloidin label. They then document »benefits« of the Fhos KD in terms of lifetime but also using olfactory conditioning memory testing (1 hr aversive memory) as well as assays of gut barrier integrity and locomotor activity. They then monitor autophagy in their context, using ATG8 immunostainings as well as a tandem GFP-mCherry-ATG8a sensor. They interpret their findings as loss of F-actin restoring brain/neuron autophagy. Next, they address mitochondrial status and the mitophagic situation, documenting an increase of »mito-lysosomes« and a decrease of ATP5a upon pan-neuronal Fhos KD. Finally, they combine KD of ATG1 with their Fhos KD. While the reversal of the age induced F-actin rod accumulation is still present here, neither the longevity improvement nor the decrease of ATP5A label in aged animals after Fhos KD are anymore observed. Overall: I do find their most important finding of reversing the F-actin accumulation in aging *Drosophila* (brains) to be »protective« very interesting, so that the manuscript in principle could be a candidate for Nature Comm.

Thanks for recognising the importance of our study and the appropriateness for this journal. We sincerely appreciate your careful and thorough evaluation of our manuscript.

This said, I am somewhat less convinced by their data stating that a restauration of brain autophagy/mitophagy is necessarily responsible here. This also as any mechanistic suggestion of how this crosstalk (F-actin with (mito)autophagy) would operate is missing. In this regard, at least some effort concerning the sub-cellular nature of the F-actin accumulation would have been appropriate. Moreover, it do have some concerns with some of the autophagy/mitophagy-related data and their interpretation.

Points:

- Concerning the autophagy assays: is an accumulation of ATG8 puncta really accepted to be an indication of their being autophagy defects?

We (Schmid et al Nature Aging, 2022; Aparicio et al, Cell Reports 2019) and others (<https://pubmed.ncbi.nlm.nih.gov/29806784/>) have previously reported that an accumulation of ATG8 puncta is linked to impaired autophagy in aged tissues. However, we do not rely exclusively on this readout. As noted, we also used a reporter line expressing GFP-mCherry-ATG8a (“ATG8a-tandem”) ubiquitously under the control of the endogenous ATG8 promoter. Furthermore, we also examined the levels of the autophagy adaptor protein p62. It is widely accepted that impaired autophagy is associated with increased levels of p62 in mammals and *Drosophila* (<https://pubmed.ncbi.nlm.nih.gov/33634751/>). All of these assays that we employed validate and support the finding that actin dysregulation in the aged brain leads to disabled autophagy. We believe that using multiple readouts of autophagy strengthens our conclusions. Moreover, we have examined autophagy in response to both genetic interventions and pharmacological interventions that reduce actin polymerization in the aged brain. In this revision, we now include data showing that both genetic and pharmacological interventions that lead to excessive levels of F-actin produce disabled brain autophagy. Hence, we believe that our findings are robust.

Related: given that they find more ATG8 puncta in the anti-ATG8 immunofluorescence staining (Fig. 4A), shouldn't this also speak up when using the tandem ATG8 GFP/mCherry sensor, as the sum/total integral of mCherry/GFP signals being increased in aged brains? I did find making the »red-only area« the only parameter shown a somewhat selective. How about the red/green ratios? It is a game of emphasizing total versus relative amounts of autophagic vacuole (maturation), but a normalization of mCherry-only spots to total mCherry-GFP-signal would be informative here.

Thank you for this suggestion. We have included additional analyses of ATG8-tandem to include comparisons of the ratio of mCherry-only puncta to total mCherry+GFP signal across conditions below. Assessing mCherry-only puncta indicates low pH-driven quenching of GFP signal of the mCherry-GFP-ATG8 fusion protein, and comparing mCherry-only signal to total mCherry-GFP signal, as suggested, validates our observations in changes to autophagic flux while taking into account the observed differences in overall signal of the tandem fusion protein in aged brains.

- For the »mito-lysosomes« (Fig. 4A): isn't there also more GFP signals in the aged Fhos KD, at least their images suggest this? How about the ratio GFP/Cherry ratio here? From what I do see I am afraid I cannot agree on there being more (means a higher ratio) mitochondria in lysosomes (side note, knowing that previous papers indeed : the term »mitolysosome« in my ears is somewhat confusing as it implies lysosomes within/at mitochondria not vice versa, though I do know it was used in the literature in some papers like that). Fig. 4A: why are no young control animals shown?

Thank you for your observations and insightful comments. We have now included the young control animals in mitoQC panels. Below, we have also included comparisons of the ratio of mCherry to GFP area from each mitoQC experiment. We find that this helps to further validate our conclusion that knocking down *Fhos* in neurons improves the amount of mitophagy taking place in aged brains as indicated by both the amount of mCherry-only puncta (as previously described), as well as the overall shift in more mCherry signal compared to GFP. Both of these readouts in aged brains with neuronal *Fhos* knockdown more closely resemble young control brains compared to control aged brains. We would also like to add that this is not the only readout of mitophagy that we examined. We show improved TMRE staining (Figure 4e,f), reduced mitochondrial content (Figure 4c,d) and a genetic requirement for autophagy (Figures 3,5 and Supplementary Fig. 3a,b).

elavGS>UAS-mitoQC
UAS-Fhos-RNAi

elavGS>UAS-mitoQC

- Their cellular/cell biological description of the F-actin rods (definition of rods?) is very shallow/essentially missing. Is there an accumulation in neuronal cell bodies, axons and dendrites (or at least in the »neuropile«) to the same extent? Are their rods displaced as compared to the physiological localization of F-actin? And: where in the brain are their images (Fig. 2c ...) taken from? Always the same ?

We agree that our original analysis of the phenomenon lacked some important information. Hence, we are grateful for this excellent comment/question. We have now expanded upon our original observations and characterizations of F-actin accumulation in aging brains, primarily in Supplementary Fig. 1. We have found localization of the F-actin-rich rods in the neuropil of the optic lobe as indicated by an anti-N-cadherin antibody (Supplementary Fig 1h). This synapse-rich region of the *Drosophila* brain has been well-characterized to largely consist of axons and dendrites. In contrast, we did not observe changes to F-actin accumulation in neuronal cell bodies in the optic lobe. Furthermore, to further define the age-related accumulation of brain F-actin, we have expanded our investigation to include other major cell populations in the brain, primarily glia. We tested for age-associated F-actin changes in this region using neuron-specific versus glia-specific GAL4 lines crossed with mCD8-GFP. While we found striking signal and overlap with F-actin and neurons, we found little or no localization with reporters for perineural, subperineural, ensheathing, cortex, or astrocyte-like glia (Supplementary Fig. 1i). Additionally, we investigated other well-defined regions of the *Drosophila* midbrain, specifically the antennal lobe and mushroom body. While we failed to detect a significant change to F-actin in the mushroom body (Supplementary Fig. 2c,d), we did see an age-associated increase in F-actin in the antennal lobe (Supplementary Fig. 2a,b). To improve clarity on where these images were taken, we have now included diagrams of the *Drosophila* brain next to imaging data with circles indicating respective locations.

- Instead of using here »young/old« and at other places specifying the exacte age (e.g. 21 d), I suggest they should always give the exacte age of the flies, as the exacte age is relevant for interpretation. It looks as if the choice of analysis time points was somewhat arbitray (Fig. 1g,h,i vs Fig. J,k,l)

Done

- They did ELISA from head homogenates of young vs old flies and present an actin-F accumulation (from what I do see there was no normalization to protein amounts, just to fly head number, correct? Would be interesting to see total protein normalized data; also: analyzing dissected brains instead of heads would be more appropriate in my eyes). Same time, they observe an upregulation of actin mRNA. Isn't that a likely source of the increased F-actin label? Is the transcriptional increase a rather special feature, or

is this found for a majority of analyzed transcripts in an RNAseq experiment? How about testing heterozygosity of an actin locus to suppress aspects of aging?

Thank you for these points of clarification. We agree that dissected brains to assess protein levels (versus total heads) would have been preferred, but due to technical challenges with handling brains in isolation buffer, we found greater consistency in each condition when handling entire heads. For the ELISA of head homogenates of young vs. old flies, we normalized the total amount of protein by the fly head number and validated by nanodrop. We have also performed a Bradford assay to compare total protein levels in the head homogenates across the age groups (Supplementary Fig. 1g). In contrast to the changes to F-actin levels in heads with age, we found consistent levels of total protein. We agree that the observed increase in *Act5c* and *Act42a* transcripts are likely associated with the downstream increase in levels of actin protein. We agree that RNA sequencing would be an excellent complement to this investigation, but it is beyond the scope of this current study. We also agree with the importance of testing if the capacity to offset accumulation of brain F-actin (transcripts and protein) by reducing cytoplasmic actin isoforms directly. To do so, we tested neuronal specific knockdown of *Act5c* and *Act42a* (Supplementary Fig. 2l,m). Midlife neuronal expression of *Act5c-RNAi* significantly extended longevity. Conversely, overexpression of *Act5c* and *Act42a* in neurons shortened lifespan (Supplementary Fig. 2p,q). We further assessed the effect of neuronal *Act5c* knockdown by imaging to assess changes to F-actin (Supplementary Fig. 2r,s) and brain mitochondrial content (Supplementary Fig. 2j,k).

- There is a body of *Drosophila* literature concerning the age-induced decline of olfactory memory scores, the role of autophagy herein, and the protective effects of dietary restriction mimetics. I think it would be appropriate to cite at least some of this work.

We agree and have now cited this work

Along the same lines: what they apparently do measure for my understanding is one hour aversive mid-term olfactory memory, I think they should stay with normally used nomenclature here, again they are somewhat disregarding previous work. Per se, there would be the possibility to separate anesthesia sensitive from resistant by post.-training amnesic cooling, means to separate ASM from ARM. So far, it was ASM found to be age-sensitive. They thus could better connect their data to existing literature.

We did not anesthetize the flies for the olfactory training assay. We agree that it would be interesting to further dissect in a more detailed manner the precise forms of olfactory function and memory that are impacted by altered actin dynamics during aging. Indeed, moving forward, this is something that we're interested in pursuing. However, we consider this to be beyond the scope of this initial manuscript.

- Fig.2g vs 5g: why is the absolute lifetime of the two controls that different? Looks like 50 (2g) vs 75 (5g) days LD50. The *elavGS>UAS-ATG1-RNAi,UAS-Fhos-RNAi* seem to have the same lifetime as the *elavGS> UAS-Fhos-RNAi*. I am prepared to accept that absolute lifetimes measured are often somewhat variant, however, this difference between controls worries me that the effect is not necessarily due to the KD of ATG1 here...

We understand what the reviewer is saying and we note the difference also. This is the result of generating new lines for epistasis analysis of this kind. We believe that it is preferable and more meaningful to not cross transgenes into an inbred background.

- There were many labeling mistakes etc..care should ve taken in case of a revision..

Thanks for this. We have rectified any labeling errors.

- There is a previous paper (Anhezini, L., Saita, A.P., Costa, M.S., Ramos, R.G., Simon, C.R. (2012). Fhos encodes a Drosophila Formin-Like Protein participating in autophagic programmed cell death. *genesis* 50(9): 672—684), speaking about the nuclear transfer of Formin (and more literature reporting nuclear functions of Formins). How sure can they be that their formin KD longevity phenotype really is due to the loss of F-actin (I do see that they used a pharmacological assay as well, however, they report less age-protection findings here)? How about the results of their actin modulator mini-screen..any more supportive evidence from this approach?

Yes, we show that numerous manipulations of actin dynamics impacts aspects of tissue and organismal aging. In addition to Fhos RNAi, we now show in Supplementary Fig. 2l,m,n,o the effects of actin isoform knockdown as well as the overexpression of genes promoting actin filament turnover (cofilin and gelsolin) on lifespan. In this figure we have also found a comparable effect in abrogating F-actin rods in aged brains by neuronal induction of *Act5c-RNAi* (Supplementary Fig. 2r,s). As noted, we also show that pharmacological disruption of F-actin improves cognition (Figure 2d) and restores mitochondrial homeostasis (Figure 2g,h,i,j and Supplementary Fig. 2a,b,q,r,s,t) and autophagy (Figure 3g,h,i,j) in the aged brain!!! Finally, we now include data showing that both genetic and pharmacological treatments that *increase F-actin* levels confer disabled brain

autophagy (Supplementary Fig. 3c,d,e,f,g)! Hence, we are very confident in our robust findings.

Reviewer #3 (Remarks to the Author):

In this manuscript, Schmid and colleagues examined the role of actin dynamics in the aging brain using *Drosophila* model. They find a great increase of filamentous actin in aging fly brains. Through genetic and pharmacological manipulation of actin polymerization, the authors conclude that excess actin polymerization disrupts neuronal homeostasis (such as autophagy, mitophagy, mitochondrial function), while neuronal knockdown of polymerization factor Fhos extends lifespan and preserves neuronal function with age. Thus, the work uncovers a novel interplay between actin polymerization and brain aging.

Thanks for recognizing the novelty of our work. We sincerely appreciate your careful and thorough evaluation of our manuscript.

I have a few concerns as listed below.

1. The lifespan extending effect of neuronal knockdown of Fhos is a very interesting finding. The authors mentioned that they also tested several other actin polymerization factors. I suggest the authors include these results in the paper. It is important to demonstrate whether the lifespan is regulated by specific (not general) actin polymerization factors.

Thanks for recognizing that our findings are “very interesting”. We agree that it is important to show the findings with other actin polymerization factors. In Supplementary Fig. 2l,m,n,o, we show lifespans of flies with neuronal knockdown of the actin isoforms expressed by neurons (*Act5c* and *Act42a*), as well as overexpression of two genes associated with actin filament disassembly, *Twinstar/cofilin* and *Gelsolin*. Conversely, in this figure we also show shortening of lifespan associated with overexpressing *Act5c* and *Act42a* in neurons (Supplementary Fig. 2p,q). Given the diverse but complementary function of these approaches and genes, we feel this strengthens the broader claim of targeting disrupted actin dynamics in aging brains to modulate organismal lifespan.

2. There is no justification for why the pan neuronal driver is used in the genetic manipulations of polymerization factors. The authors show that actin increases in the aging brain. However, it is unclear whether this increase mainly occurs in neurons or other cell types of the central brain, such as glial cells. The authors need to demonstrate whether the elevated actin polymerization is found in aging neuronal tissue, and/or glial

cells. It would be great if they could also test if neuronal knockdown of Fhos attenuates age-associated actin rods in neuronal tissues or glial cells using cell type-specific markers.

This is a fantastic question, which we have now explored in some detail. As a result, we now incorporate a significant body of new data, which we believe not only strengthens our findings but also greatly improves our manuscript. Thanks for helping us improve our manuscript.

In the first place, in Supplementary Fig. 2k, we report that glial-specific knockdown of Fhos does NOT extend lifespan. We have also now examined whether the elevated actin polymerization is found in neurons and/or glia during aging. In Supplementary Fig. 1i, we expressed a GFP reporter via GAL4 drivers specific for neurons or glial populations, specifically perineural, subperineural, cortex, astrocyte-like, and ensheathing glia. We found striking signal and overlap between the neuronal reporter and the F-actin-rich rods associated with age, and little or no localization of this F-actin with reporters for glia. We feel that this complements and strengthens our approach to targeting age-associated F-actin accumulation in brains primarily through the use of neuronal drivers.

It is clear that actin contents increase in the aging brain. But how exactly aging alters actin contents was not fully addressed. Even though the results from Fhos knockdown experiments suggest a possible involvement of actin polymerization, the age-related change of actin structure is so dramatic and it is highly likely the remodeling of neuronal structure (e.g., axonal remodeling) is another underlying cause. The authors should discuss this possibility in the manuscript.

Thanks for this suggestion. We have added this to the discussion.

3. It appears that most of the imaging analysis was focused on the optic lobe, in particular the medulla layer. I wonder if the changes of age-associated actin rods are also prominent in other regions of the brain, for example, antennal lobe and mushroom bodies. The reason I am asking this is that the authors did a learning and memory test using olfaction aversion training, which is associated with the function of the antennal lobe and mushroom bodies. Can the authors check if antennal lobe and mushroom bodies also exhibit age-associated increases in actin rods, and if neuronal knockdown of Fhos could attenuate it?

This is another great suggestion! Thanks! We have now done so. As shown in Supplementary Fig. 2a,b we expanded our survey of the aging brain to show that F-actin indeed increases with age in the antennal lobe and can be reduced to levels comparable to young controls with neuronal knockdown of *Fhos*. However, we did not observe a change to F-actin levels in the mushroom body with age or *Fhos* knockdown (Supplementary Fig. 2c,d). It would be interesting to investigate how these age-related changes to F-actin might affect these brain regions, such as olfactory function and memory, in future studies.

As a side note, the authors should clarify in the text or figure legend which regions of the brain were used in each fluorescence image, including all actin staining, autophagy, mitophagy, and mitochondrial images.

Thanks for this helpful suggestion. We have now done so.

4. It is a little bit counterintuitive that inhibition of actin polymerization can reverse age-related decline in autophagic flux. As mentioned by the authors, actin dynamics are essential in the biogenesis and transportation of most cellular vesicles, including autophagosomes. To further demonstrate that excess actin polymerization is detrimental and impairs autophagic activity, the authors could genetically enhance actin polymerization or treat flies with actin polymerizing and stabilizing drugs and then test if excess actin polymerization blocks autophagy and mitophagy in the brain.

We agree that this is interesting. We have now done so. As shown in Supplementary Fig. 3c,d, we find that neuronal overexpression of *Act42a* results in early-onset marker of stalled autophagy/mitophagy (p62 accumulation). Furthermore, pharmacological stabilization of actin using jasplakinolide resulted in the formation of early brain F-actin-rich rods and disabled autophagy (Supplementary Fig. 3e,f,g). Together, we believe that these results strengthen our interpretation that excess F-actin polymerization impairs autophagy in the brain.

5. The authors used an anti-dsDNA antibody to visualize mtDNA. How specific is this antibody? Have the authors verified its specificity experimentally (to exclude the cross-reaction with nuclear dsDNA)?

We understand why the reviewer would ask this question. Unfortunately, the ds-DNA antibody is NOT specific to mtDNA. Hence, we now label accordingly as dsDNA in the relevant figure.

6. Minor issues:

Fig 4a,b young age data are needed. Why is the GFP-positive structure ring shape, but the mCherry structure exhibits puncta shape?

We have added the young age data. The question of the shape of the GFP/mCherry puncta is very interesting!! We could speculate, but it would be pure speculation. We don't think anyone can provide a definitive answer here. There is much to learn.

Line 293, "Supplementary Fig 3c,d" should be "Supplementary Fig 4c,d"

Done

Line 295, "Supplementary Fig e,f" missing figure number

Done

REVIEWERS' COMMENTS

Reviewer #1 (Remarks to the Author):

This is a revised manuscript from David Walker's research group describing the exciting observation that excess actin polymerization is a novel biomarker for brain aging. By means of a preponderance of evidence, they show that brain aging in *Drosophila* is associated with an accumulation of filamentous actin (F-actin) and that this can be suppressed by pro-longevity cues. They further define the crucial role autophagy plays in this process. A major strength of the study is the fact that it is the first to link actin dynamics to brain aging. Furthermore, the multiple genetic tools and reagents used would be directly applicable to studying actin dynamics in other organs such as the muscle and gut. For instance, Act88F regulates indirect flight muscle formation and may have a muscle-specific role; the relevance of this gene to muscle aging can now be explored using techniques analogous to what is described in this paper. Similarly, the relevance of other actin isoforms such as Act57B, Act79B, Act87E and Activin-beta to aging can also now be explored. In addition, different isoforms of the actin cross-linking protein, alpha-actinin, are expressed in the muscle, ring canal of the ovary, etc. – the significance of these differences to actin dynamics and aging of the relevant tissues can now be investigated. Actin also has many posttranslational modifications, including acetylation, phosphorylation, arginylation, glutathionylation, etc. – which can now be explored for possible roles in *Drosophila*. Thus, this manuscript is highly significant as it describes a “paradigm-shifting” study that will provide an opportunity to answer many questions about aging, healthspan and actin dynamics in multiple *Drosophila* tissues. The few concerns I had have been thoroughly addressed, and concerns raised by the other two reviewers have been resolved as well. Accordingly, I recommend that this manuscript be accepted for publication in *Nature Communications*. I look forward to seeing this exciting manuscript in press.

Reviewer #2 (Remarks to the Author):

The authors went through a truly extensive revision, including the addition of substantial new data. While per se there would still be room for a deeper analysis, given the novelty and potential importance of these findings, I do suggest publication of the paper in its current form.

Reviewer #3 (Remarks to the Author):

The authors have fully addressed my previous comments. They have done a great deal of new experiments in the revision, and many of which offer important insights into the interplay between actin polymerization and brain aging. I recommend it for publication in Nature Communications.